# Are you using test log-likelihood correctly?

**Sameer K. Deshpande**[*]  *sameer.deshpande@wisc.edu*
*University of Wisconsin–Madison*

**Soumya Ghosh**[*]  *ghoshso@us.ibm.com*
*MIT-IBM Watson AI Lab*
*IBM Research*

**Tin D. Nguyen**[*]  *tdn@mit.edu*
*MIT-IBM Watson AI Lab*
*Massachusetts Institute of Technology*

**Tamara Broderick**  *tbroderick@mit.edu*
*MIT-IBM Watson AI Lab*
*Massachusetts Institute of Technology*

**Reviewed on OpenReview:** *https://openreview.net/forum?id=n2YifD4Dxo*

## Abstract

Test log-likelihood is commonly used to compare different models of the same data or different approximate inference algorithms for fitting the same probabilistic model. We present simple examples demonstrating how comparisons based on test log-likelihood can contradict comparisons according to other objectives. Specifically, our examples show that (i) approximate Bayesian inference algorithms that attain higher test log-likelihoods need not also yield more accurate posterior approximations and (ii) conclusions about forecast accuracy based on test log-likelihood comparisons may not agree with conclusions based on root mean squared error.

## 1 Introduction

Test log-likelihood, also known as predictive log-likelihood or test log-predictive, is computed as the log-predictive density averaged over a set of held-out data. It is often used to compare different models of the same data or to compare different algorithms used to fit the same probabilistic model. Although there are compelling reasons for this practice (Section 2.1), we provide examples that falsify the following, usually implicit, claims:

- **Claim**: The higher the test log-likelihood, the more accurately an approximate inference algorithm recovers the Bayesian posterior distribution of latent model parameters (Section 3).

- **Claim**: The higher the test log-likelihood, the better the predictive performance on held-out data according to other measurements, like root mean squared error (Section 4).

Our examples demonstrate that test log-likelihood is not always a good proxy for posterior approximation error. They further demonstrate that forecast evaluations based on test log-likelihood may not agree with forecast evaluations based on root mean squared error.

We are not the first to highlight discrepancies between test log-likelihood and other analysis objectives. For instance, Quiñonero-Candela et al. (2005) and Kohonen & Suomela (2005) showed that when predicting

---

[*]These authors contributed equally to this work.

discrete data with continuous distributions, test log-likelihood can be made arbitrarily large by concentrating probability into vanishingly small intervals. Chang et al. (2009) observed that topic models with larger test log-predictive densities can be less interpretable. Yao et al. (2019) highlighted the disconnect between test log-likelihood and posterior approximation error in the context of Bayesian neural networks. Our examples, however, reveal more fundamental discrepancies between test log-likelihood and other evaluation metrics. In particular, we show how comparisons based on test log-likelihood can contradict comparisons based on other objectives even in simple models like linear regression.

After introducing our notation, we precisely define test log-likelihood and review arguments for its use in Section 2. In Sections 3.1–3.3, we present several examples showing that across a range of posterior approximations, those with higher test log-likelihoods may nevertheless provide worse approximation quality. Then, in Section 3.4, we provide some intuition about why this phenomenon can occur when the model is severely misspecified (Section 3.1); when using sophisticated posterior approximation methods (Section 3.2); and even when there is little or no model misspecification (Section 3.3). In Section 4, we show examples in both complex and simple models where test log-likelihood is higher but root mean squared error on held-out data is worse. Our examples in Section 4 do depend on model misspecification, but we note that model misspecification is unavoidable in practice. We conclude in Section 5 with a reflection on when we should use test log-likelihood in practice.

## 2 Background

We assume we have access to training and testing data such that all data points are independently and identically distributed (i.i.d.) from an unknown probability distribution $\mathcal{P}$. Let $\mathcal{D} = \{y_n\}_{n=1}^N$ denote the training data. In many standard analyses, practitioners will have access to a predictive density of a future data point $y^\star$ given the observed $\mathcal{D}$: $\pi(y^\star|\mathcal{D})$. For instance, consider the following three cases.

- Case A: Practitioners often model the observed data by introducing a parameter $\theta$ and specifying that the data are i.i.d. from a conditional distribution $\Pi(Y|\theta)$ with density $\pi(y|\theta)$. In a non-Bayesian analysis, one usually computes a point estimate $\hat{\theta}$ of the unknown parameter (e.g. by maximum likelihood). Given a point estimate $\hat{\theta}$, the predictive density $\pi(y^\star|\mathcal{D})$ is just $\pi(y^\star|\hat{\theta})$.

- Case B: A Bayesian analysis elaborates the conditional model from Case A by specifying a prior distribution $\Pi(\theta)$ and formally computes the density $\pi(\theta|\mathcal{D})$ of the posterior distribution $\Pi(\theta|\mathcal{D})$ from the assumed joint distribution $\Pi(\mathcal{D}, \theta)$. The Bayesian posterior predictive density is given by

$$\pi(y^\star|\mathcal{D}) = \int \pi(y^\star|\theta)\pi(\theta|\mathcal{D})d\theta. \tag{1}$$

- Case C: An approximate Bayesian analysis proceeds as in Case B but uses an approximation in place of the exact posterior. If we let $\Pi(\theta|\mathcal{D})$ represent an approximation to the exact posterior, Equation (1) yields the approximate Bayesian posterior predictive density $\pi(y^\star|\mathcal{D})$. Sometimes, due the difficulty of the integral in Equation (1), a further approximation may be used to yield a predictive density $\pi(y^\star|\mathcal{D})$.

In all of these cases, we will refer to the practitioner as having access to a model $\Pi$ that determines the predictive distribution $\Pi(y^\star|\mathcal{D})$; in particular, we allow "model" henceforth to encompass fitted models and posterior approximations. One can ask how well the resulting $\Pi(y^\star|\mathcal{D})$ predicts new data generated from $\mathcal{P}$. Practitioners commonly assess how well their model predicts out-of-sample using a held-out set of testing data $\mathcal{D}^\star = \{y_n^\star\}_{n=1}^{N^\star}$, which was not used to train the model. To compute test log-likelihood, they average evaluations of the log-predictive density function over the testing set:

$$\text{TLL}(\mathcal{D}^\star; \Pi) := \frac{1}{N^\star} \sum_{n=1}^{N^\star} \log \pi(y_n^\star|\mathcal{D}), \tag{2}$$

where our notation makes explicit the dependence of the test log-likelihood (TLL) on testing data $\mathcal{D}^\star$ and the chosen model $\Pi$. In particular, researchers commonly use test log-likelihood to select between two models of

the data, say $\Pi$ and $\tilde{\Pi}$; that is, they select model $\Pi$ over $\tilde{\Pi}$ whenever $\text{TLL}(\mathcal{D}^\star; \Pi)$ is higher than $\text{TLL}(\mathcal{D}^\star; \tilde{\Pi})$. Note that the abbreviation NLPD (negative log predictive density) is also commonly used in the literature for the negative TLL (Quiñonero-Candela et al., 2005; Kohonen & Suomela, 2005). In Appendix C, we briefly discuss some alternative metrics for model comparison.

## 2.1 The case for test log-likelihood

In what follows, we first observe that, if we wanted to choose a model whose predictive distribution is closer to the true data distribution in a certain KL sense, then it is equivalent to choose a model with higher *expected log-predictive density* (elpd). Second, we observe that TLL is a natural estimator of elpd when we have access to a finite dataset.

**The unrealistic case where the true data-generating distribution is known.** The expected log-predictive density is defined as

$$\text{elpd}(\Pi) := \int \log \pi(y^\star | \mathcal{D}) d\mathcal{P}(y^\star).$$

Our use of the abbreviation elpd follows the example of Gelman et al. (2014, Equation 1). If we ignore an additive constant not depending on $\Pi$, $\text{elpd}(\Pi)$ is equal to the negative Kullback–Leibler divergence from the predictive distribution $\Pi(y^\star | \mathcal{D})$ to the true data distribution $\mathcal{P}(y^\star)$. Specifically, if we assume $\mathcal{P}$ has density $p(y^\star)$, we have

$$\text{KL}\left(\mathcal{P}(y^\star) \,\|\, \Pi(y^\star | \mathcal{D})\right) = \int p(y^\star) \log p(y^\star) dy^\star - \text{elpd}(\Pi).$$

Thus, $\text{elpd}(\Pi) > \text{elpd}(\tilde{\Pi})$ if and only if the predictive distribution $\Pi(y^\star | \mathcal{D})$ is closer, in a specific KL sense, to the true data distribution than the predictive distribution $\tilde{\Pi}(y^\star | \mathcal{D})$ is.

**Test log-likelihood as an estimator.** Since we generally do not know the true generating distribution $\mathcal{P}$, computing $\text{elpd}(\Pi)$ exactly is not possible. By assumption, though, the test data are i.i.d. draws from $\mathcal{P}$. So $\text{TLL}(\mathcal{D}^\star; \Pi)$ is a computable Monte Carlo estimate of $\text{elpd}(\Pi)$. If we assume $\text{elpd}(\Pi)$ is finite, it follows that a Strong Law of Large Numbers applies: as $N^\star \to \infty$, $\text{TLL}(\mathcal{D}^\star; \Pi)$ converges almost surely to $\text{elpd}(\Pi)$. Therefore, with a sufficiently high amount of testing data, we might compare the estimates $\text{TLL}(\mathcal{D}^\star; \Pi)$ and $\text{TLL}(\mathcal{D}^\star; \tilde{\Pi})$ in place of the desired comparison of $\text{elpd}(\Pi)$ and $\text{elpd}(\tilde{\Pi})$. Note that the Strong Law follows from the assumption that the $y_n^\star$ values are i.i.d. under $\mathcal{P}$; it does not require any assumption on the model $\Pi$ and holds even when the model $\Pi$ is misspecified.

## 2.2 Practical concerns

Since $\text{TLL}(\mathcal{D}^\star; \Pi)$ is an estimate of $\text{elpd}(\Pi)$, it is subject to sampling variability, and a careful comparison would ideally take this sampling variability into account. We first elaborate on the problem and then describe one option for estimating and using the sampling variability in practice; we take this approach in our experiments below.

To start, suppose we had another set of $N^\star$ testing data points, $\tilde{\mathcal{D}}^\star$. Then generally $\text{TLL}(\mathcal{D}^\star; \Pi) \neq \text{TLL}(\tilde{\mathcal{D}}^\star; \Pi)$. So it is possible, in principle, to draw different conclusions using the TLL based on different testing datasets. We can more reasonably express confidence that $\text{elpd}(\Pi)$ is larger than $\text{elpd}(\tilde{\Pi})$ if the lower bound of a confidence interval for $\text{elpd}(\Pi)$ exceeds the upper bound of a confidence interval for $\text{elpd}(\tilde{\Pi})$.

We next describe one way to estimate useful confidence intervals. To do so, we make the additional (mild) assumption that

$$\sigma_{\text{TLL}}^2(\Pi) := \int \left[\log \pi(y^\star | \mathcal{D}) - \text{elpd}(\Pi)\right]^2 d\mathcal{P}(y^\star) < \infty.$$

Then, since the $y_n^\star$ are i.i.d. draws from $\mathcal{P}$, a Central Limit Theorem applies: as $N^\star \to \infty$,

$$\sqrt{N^\star}\left(\text{TLL}(\mathcal{D}^\star; \Pi) - \text{elpd}(\Pi)\right) \xrightarrow{\text{d}} \mathcal{N}(0, \sigma_{\text{TLL}}^2(\Pi)).$$

Although we cannot generally compute $\sigma_{\text{TLL}}(\Pi)$, we can estimate it with the sample standard deviation $\hat{\sigma}_{\text{TLL}}(\Pi)$ of the evaluations $\{\log \pi(y_n^\star | \mathcal{D})\}_{n=1}^{N^\star}$. The resulting approximate 95% confidence interval for $\text{elpd}(\Pi)$

is $\mathrm{TLL}(\mathcal{D}^\star; \Pi) \pm 2\hat{\sigma}_{\mathrm{TLL}}/\sqrt{N^\star}$. In what follows, then, we will conclude $\mathrm{elpd}(\Pi) > \mathrm{elpd}(\tilde{\Pi})$ if

$$\mathrm{TLL}(\mathcal{D}^\star; \Pi) - 2\hat{\sigma}_{\mathrm{TLL}}(\Pi)/\sqrt{N^\star} > \mathrm{TLL}(\mathcal{D}^\star; \tilde{\Pi}) + 2\hat{\sigma}_{\mathrm{TLL}}(\tilde{\Pi})/\sqrt{N^\star}. \tag{3}$$

For the sake of brevity, we will still write $\mathrm{TLL}(\mathcal{D}^\star; \Pi) > \mathrm{TLL}(\mathcal{D}^\star; \tilde{\Pi})$ in place of Equation (3) below.

To summarize: for a sufficiently large test dataset $\mathcal{D}^\star$, we expect predictions made from a model with larger TLL to be closer (in the KL sense above) to realizations from the true data-generating process. In our experiments below, we choose large test datasets so that we expect TLL comparisons to reflect elpd comparisons. Our experiments instead illustrate that closeness between $\Pi(y^\star|\mathcal{D})$ and $\mathcal{P}$ (in the KL sense above) often does not align with a different stated objective.

# 3 Claim: higher test log-likelihood corresponds to better posterior approximation

In this section, we give examples where test log-likelihood is higher though the (approximation) quality of an approximate posterior mean, variance, or other common summary is lower. We start with examples in misspecified models and then give a correctly specified example. We conclude with a discussion of the source of the discrepancy: even in the well-specified case, the Bayesian posterior predictive need not be close to the true data-generating distribution.

Practitioners often use posterior expectations to summarize the relationship between a covariate and a response. For instance, the posterior mean serves as a point estimate, and the posterior standard deviation quantifies uncertainty. However, as the posterior density $\pi(\theta|\mathcal{D})$ is analytically intractable, practitioners must instead rely on approximate posterior computations. There are myriad approximate inference algorithms – e.g. Laplace approximation, Hamiltonian Monte Carlo (HMC), mean-field variational inference, to name just a few. All these algorithms aim to approximate the same posterior $\Pi(\theta|\mathcal{D})$. Test log-likelihood is often used to compare the quality of different approximations, with higher TLL values assumed to reflect more accurate approximations, e.g. in the context of variational inference (see, e.g., Hoffman et al., 2013; Ranganath et al., 2014; Hernández-Lobato et al., 2016; Liu & Wang, 2016; Shi et al., 2018) or Bayesian deep learning (see, e.g., Hernández-Lobato & Adams, 2015; Gan et al., 2016; Li et al., 2016; Louizos & Welling, 2016; Sun et al., 2017; Ghosh et al., 2018; Mishkin et al., 2018; Wu et al., 2019; Izmailov et al., 2020; 2021; Ober & Aitchison, 2021).

Formally, suppose that our exact posterior is $\Pi(\theta|\mathcal{D})$ and that we have two approximate inference algorithms that produce two approximate posteriors, respectively $\hat{\Pi}_1(\theta|\mathcal{D})$ and $\hat{\Pi}_2(\theta|\mathcal{D})$. The exact posterior and its approximations respectively induce predictive distributions $\Pi(y^\star|\mathcal{D}), \hat{\Pi}_1(y^\star|\mathcal{D})$, and $\hat{\Pi}_2(y^\star|\mathcal{D})$. For instance, $\hat{\Pi}_1(\theta|\mathcal{D})$ could be the empirical distribution of samples drawn using HMC and $\hat{\Pi}_2(\theta|\mathcal{D})$ could be a mean-field variational approximation. Our first example demonstrates that it is possible that (i) $\mathrm{TLL}(\mathcal{D}^\star; \hat{\Pi}_1) > \mathrm{TLL}(\mathcal{D}^\star; \Pi)$ but (ii) using $\hat{\Pi}_1$ could lead to different inference about model parameters than using the exact posterior $\Pi$. Our second example demonstrates that it is possible that (i) $\mathrm{TLL}(\mathcal{D}^\star; \hat{\Pi}_1) > \mathrm{TLL}(\mathcal{D}^\star; \hat{\Pi}_2)$ but (ii) $\hat{\Pi}_1(\theta|\mathcal{D})$ is a worse approximation to the exact posterior $\Pi(\theta|\mathcal{D})$ than $\hat{\Pi}_2(\theta|\mathcal{D})$.

## 3.1 TLL and downstream posterior inference

Relying on TLL for model selection can lead to different inferences than we would find by using the exact posterior. To illustrate, suppose we observe $\mathcal{D}_{100} = \{(x_n, y_n)\}_{n=1}^{100}$ drawn from the following heteroscedastic model:

$$x_n \sim \mathcal{N}(0, 1), \quad y_n \mid x_n \sim \mathcal{N}(x_n, 1 + \log(1 + \exp(x_n))). \tag{4}$$

Further suppose we model these data with a misspecified homoscedastic model:

$$\theta \sim \mathcal{N}([0, 0]^\top, [1, 0; 0, 1]), \quad y_n \mid \theta, \phi_n \sim \mathcal{N}(\theta^T \phi_n, 1), \tag{5}$$

where $\phi_n = [x_n, 1]^\top$, and $\theta = [\theta_1, \theta_2]$. Figure 1 shows the posterior mean and the 95% predictive interval of the misspecified regression line $\theta^\top \phi$ from (A) the exact Bayesian posterior; (B) the mean field variational approximation restricted to isotropic Gaussians; and (C)–(F) variational approximations with re-scaled marginal variances. Each panel includes a scatter plot of the observed data, $\mathcal{D}_{100}$. We also report the 2-Wasserstein distance between the exact posterior and each approximation and the TLL averaged over

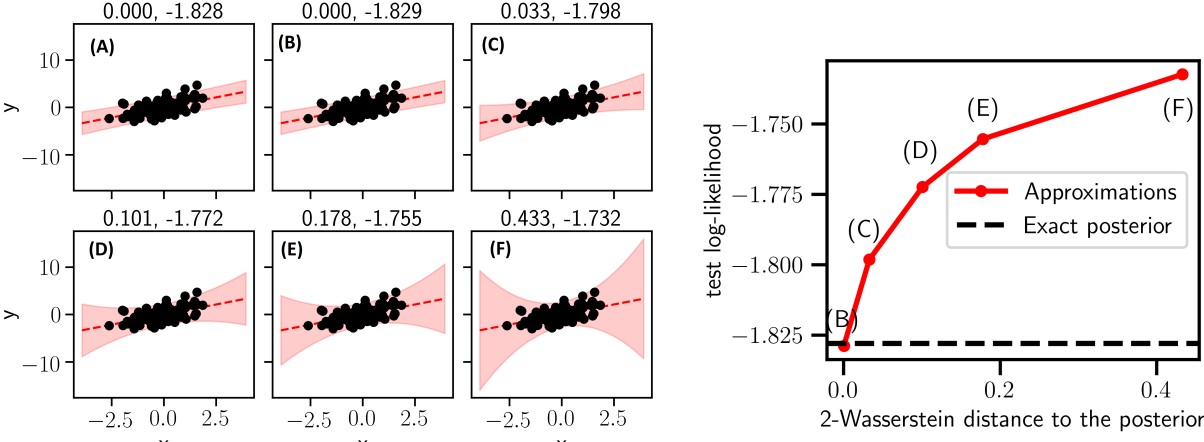

Figure 1: *(Left).* Predictive distributions under the Bayesian posterior and mean field variational approximations. The two numbers in the title of each plot are the 2-Wasserstein distance to the exact posterior and test log-likelihood computed on $10^4$ test set observations. Two standard errors in the test log-likelihood estimate are (A) 0.03, (B) 0.03, (C) 0.02, (D) 0.02, (E) 0.02, (F) 0.02. *(Right).* The relationship between 2-Wasserstein distance to the posterior and test log-likelihood.

$N^* = 10^4$ test data points drawn from Equation (4); note that the 2-Wasserstein distance can be used to bound differences in means and variances (Huggins et al., 2020). The variational approximation (panel (B) of Figure 1) is quite accurate: the 2-Wasserstein distance between the approximation and the exact posterior is $\sim 10^{-4}$. See also Figure 2, which shows the contours of the exact and approximate posterior distributions. As we scale up the variance of this approximation, we move away from the exact posterior over the parameters but the posterior predictive distribution covers more data, yielding higher TLL. The left panel of Figure 11 in Appendix B.3 shows the same pattern using the KL divergence instead of the 2-Wasserstein distance.

**TLL and a discrepancy in inferences.** Researchers are often interested in understanding whether there is a relationship between a covariate and response; a Bayesian analysis will often conclude that there is no relationship if the posterior on the corresponding effect-size parameter places substantial probability on an interval not containing zero. In our example, we wish to check whether $\theta_1 = 0$. Notice that the exact posterior distribution (panel (A) in Figures 1 and 2) is concentrated on positive $\theta_1$ values. The 95% credible interval of the exact posterior[1] is $[0.63, 1.07]$. Since the interval does not contain zero, we would infer that $\theta_1 \neq 0$. On the other hand, as the approximations become more diffuse (panels (B)–(F)), TLL increases, and the approximations begin to place non-negligible probability mass on negative $\theta_1$ values. In fact, the approximation with highest TLL (panel (F) in Figures 1 and 2) yields an approximate 95% credible interval of [-0.29,1.99], which covers zero. Had we used this approximate interval, we would have failed to conclude $\theta_1 \neq 0$. That is, in this case, we would reach a different substantive conclusion about the effect $\theta_1$ if we (i) use the exact posterior or (ii) use the approximation selected by highest TLL.

### 3.2 TLL in the wild

Next, we examine a more realistic scenario in which the difference between the quality of the posterior approximation and the exact posterior distribution TLL arises naturally, without the need to artificially increase the marginal variance of the variational approximations. To explore this situation, we will first introduce another example of misspecification and repeat the type of analysis described in Section 3.1.

---

[1] Throughout we used symmetric credible intervals formed by computing quantiles: the 95% interval is equal to the 2.5%–97.5% interquartile range.

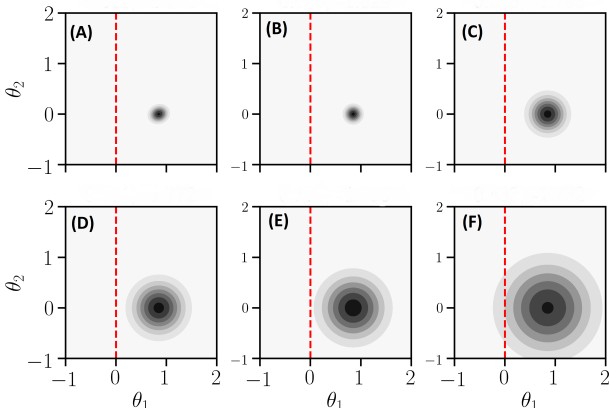

Figure 2: Contours of (A) the exact posterior, (B) the mean field variational approximation restricted to isotropic Gaussians, and (C)–(F) re-scaled mean field approximations. The line $\theta_1 = 0$ is highlighted in red.

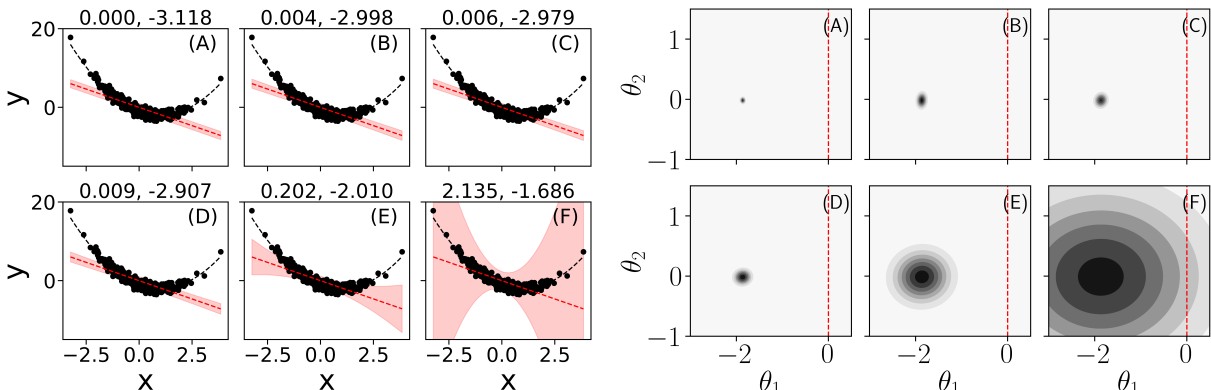

Figure 3: *(Left)*. Predictive distributions under the Bayesian posterior (A) and the SWAG posterior with SWAG learning rate of (B) $10^{-3}$, (C) $10^{-2}$, (D) $10^{-1}$, (E) 1, and (F) 10. The two numbers in the title of each plot are the 2-Wasserstein distance to the exact posterior and test log-likelihood computed on $10^4$ test set observations. Two standard errors in the test log-likelihood estimates are (A) 0.16, (B) 0.15, (C) 0.14, (D) 0.13, (E) 0.05, (F) 0.01. *(Right)*. Contours of the (A) exact posterior, and (B)–(F) SWAG approximations with different learning rates. The line $\theta_1 = 0$ is highlighted in red.

Consider the following case: we observe 500 observations $\mathcal{D}_{500} = \{(x_n, y_n)\}_{n=1}^{500}$ drawn from a non-linear model:

$$\theta_* = [-2, -1]^\top, \quad x_n \sim \mathcal{N}(0, 1), \quad y_n \mid \theta_*, \phi_n \sim \mathcal{N}(\theta_*^\top \phi_n + x_n^2, 0.5), \tag{6}$$

where $\phi_n = [x_n, 1]^\top$. Further suppose we modeled these data with a misspecified linear model

$$\theta \sim \mathcal{N}([0, 0]^\top [1, 0; 0, 1]), \quad y_n \mid \theta, \phi_n \sim \mathcal{N}(\theta^\top \phi_n, 0.5). \tag{7}$$

While the misspecification here might appear egregious, linear models are widely used in practice for modeling non-linear phenomena when one is primarily interested in inferring whether the covariates are positively correlated, negatively correlated, or are uncorrelated with the responses (Berk et al., 2014; 2018; Blanca et al., 2018; Vowels, 2023). Next, we use SWAG (Maddox et al., 2019), an off-the-shelf approximate inference algorithm, to approximate the posterior $\Pi(\theta|\mathcal{D}_{500})$. We also repeat the re-scaled variational inference experiment from Section 3.1 with this set of data and models (Equations (6) and (7)); see Appendix B.2.

SWAG uses a gradient-based optimizer with a learning rate schedule that encourages the optimizer to oscillate around the optimal solution instead of converging to it. Then, a Gaussian distribution is fit to the set of solutions explored by the optimizer around the optimum using moment matching. In general, one must select the learning rate schedule in a heuristic fashion. One might be tempted to use TLL to tune the learning rate schedule. We use this heuristic and run SWAG for a thousand epochs, annealing the learning rate down to a different constant value after 750 epochs. Although used pedagogically here, similar heuristics have been used in practice (di Langosco et al., 2022), where the learning rate is tuned based on the accuracy achieved on held-out data. We vary this constant value over the set $\{10^{-3}, 10^{-2}, 10^{-1}, 1, 10\}$. In Figure 3, we show the resulting posterior mean and the 95% predictive interval of the misspecified regression line $\theta^\top \phi$ from (A) the Bayesian posterior; (B)–(F) the SWAG posteriors using different learning rate schedules. In each plot, we overlay the observed data $\mathcal{D}_{500}$ (black dots) with the true data generating function in dashed black. We also report the 2-Wasserstein distance between the exact posterior and each approximation and the TLL averaged over $N^* = 10^4$ test data points drawn from Equation (6). In all cases, SWAG overestimates the posterior variance, with predictive distributions that better cover the data and consequently lead to a higher TLL. However, these SWAG posterior approximations are *farther* from the exact posterior. In fact, we found that a learning rate of 10 (Figure 3, *Left*, panel (F)) maximized TLL but led to the worst approximation of the exact posterior.

As in the previous section, next suppose we fit this misspecified linear model to understand whether there is a relationship between the covariates and the responses, i.e., whether $\theta_1 = 0$. Notice that the exact posterior distribution (Figure 3, *Right*, panel (A)) is concentrated on negative $\theta_1$ values, with the 95% posterior credible interval being $[-1.96, -1.79]$. Since the interval is to the left of zero, we would infer that $\theta_1 < 0$ and that the covariate and the response are negatively correlated. In contrast, if we select the SWAG approximation with the highest TLL, we select the posterior approximation in panel (F) on the right side of Figure 3. The corresponding 95% posterior credible interval is $[-4.46, 0.74]$, which places non-negligible probability mass on $\theta_1 > 0$. In this case, we would not conclude that the response and the covariate are negatively correlated – by contrast to the conclusion using the exact posterior.

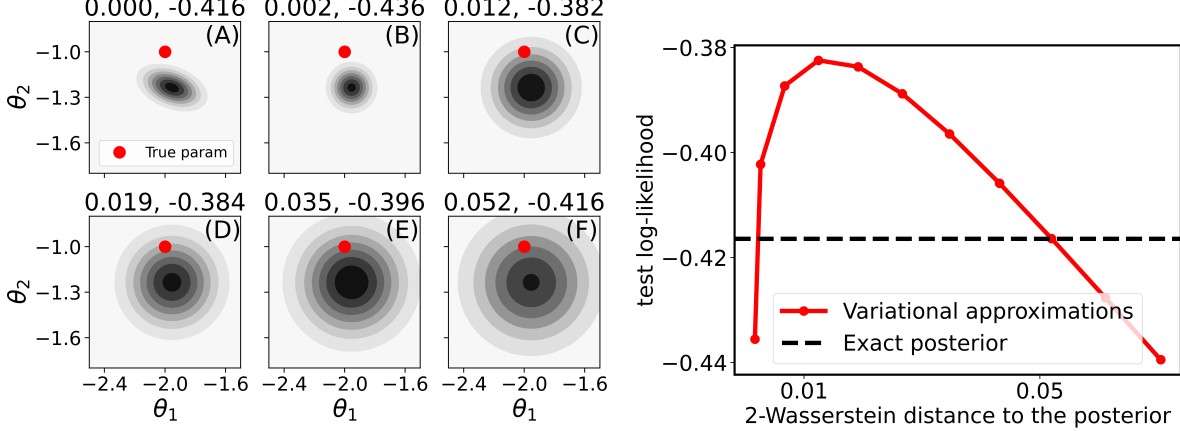

Figure 4: *(Left)*. Contours of (A) the exact posterior, (B) the mean field variational approximation restricted to isotropic Gaussians, and (C)–(F) re-scaled mean field approximations. The two numbers in the title of each plot are the 2-Wasserstein distance to the exact posterior and test log-likelihoods computed on $10^4$ test set observations. Two standard errors in the test log-likelihood estimates are (A) 0.019, (B) 0.020, (C) 0.014, (D) 0.013, (E) 0.011, (F) 0.009. *(Right)*. The non-monotonic relationship between distance to posterior and test log-likelihood. Observe that the exact posterior does not achieve highest test log-likelihood.

## 3.3 TLL and well-specified models

The examples above demonstrated that TLL is not a reliable proxy to posterior approximation quality when the model is misspecified. Though misspecified models are the norm in practice, we now demonstrate that a

distribution with higher TLL may not provide a more accurate posterior approximation even when the model is correctly specified.

To this end, consider the following Bayesian linear model:

$$\theta \sim \mathcal{N}([0,0]^\top, [1, 0.9; 0.9, 1]), \quad y_n \mid \theta, \phi_n \sim \mathcal{N}(\theta^\top \phi_n, 0.25^2), \tag{8}$$

where $\phi_n = [x_n, 1]^\top$. Now, suppose we observe ten data points $\mathcal{D}_{10} = \{(x_n, y_n)\}_{n=1}^{10}$ sampled as

$$\theta_* = [-2, -1]^\top, \quad x_n \sim \mathcal{N}(0, 1), \quad y_n \mid \theta_*, \phi_n \sim \mathcal{N}(\theta_*^\top \phi_n, 0.25^2). \tag{9}$$

The left panel of Figure 4 plots the contours of (A) the exact posterior distribution $\Pi(\phi|\mathcal{D}_{10})$; (B) the mean field variational approximation constrained to the isotropic Gaussian family; and (C)–(F) variational approximations with re-scaled marginal variances. In each panel, we report the 2-Wasserstein distance between the approximate and exact posterior and the test log-predictive averaged over $N^\star = 10^4$ test data points drawn from Equation (9).

Although we have correctly specified the conditional model of $y|(\theta, \phi)$, the exact posterior has a lower TLL than some of the approximate posteriors; in particular, the 95% confidence intervals for (C) and (D) are disjoint from the 95% confidence interval for the exact posterior, shown in (A). The left panel of Figure 4 suggests that the more probability mass an approximate posterior places around the true data-generating parameter, the higher the TLL. Eventually, as the approximation becomes more diffuse, TLL begins to decrease (Figure 4 (right)). The non-monotonicity demonstrates that an approximate posterior with larger implied TLL can in fact be further away from the exact posterior in a 2-Wasserstein sense than an approximate posterior with smaller implied TLL. The right panel of Figure 11 in Appendix B.3 demonstrates the same pattern using the KL divergence instead of the 2-Wasserstein distance. And Figure 9 in Appendix B.3 shows that, in the well-specified case, a distribution with larger TLL can provide a worse approximation of the posterior standard deviation than a distribution with smaller TLL.

### 3.4 What is going on?

We next discuss why we should not expect TLL to closely track posterior approximation quality, or posterior-predictive approximation quality. Essentially the issue is that, even in the well-specified case, the Bayesian posterior predictive distribution need not be close to the true data-generating distribution.

We illustrate these distinctions in Figure 5. The lower surface represents the space of distributions over a latent parameter $\theta$. The upper surface represents the space of distributions over an observable data point $y^\star$. Each dot in the figure represents a distribution. The two dots in the lower surface are the exact posterior $\Pi(\theta|\mathcal{D})$ (left, green dot) and an approximate posterior $\hat{\Pi}(\theta|\mathcal{D})$ (right, red dot). The three dots in the upper surface are the posterior predictive distribution $\Pi(y^\star|\mathcal{D})$ (left, green dot), the approximate posterior predictive $\hat{\Pi}(y^\star|\mathcal{D})$ (lower right, red dot), and the true data-generating distribution $\mathcal{P}(y^\star)$ (upper right, black dot). The gray lines on the left and right indicate that the distribution in the upper surface can be obtained from the corresponding (connected) distribution in the lower surface via Equation (1).

The remaining three (non-gray) lines represent three different discrepancies. Recall from Section 2 that $\mathrm{TLL}(\mathcal{D}^\star; \hat{\Pi})$ captures how close the approximate posterior predictive $\hat{\Pi}(y^\star|\mathcal{D})$ is to the true data-generating process $\mathcal{P}(y^\star)$ in a particular KL sense:

$$\mathrm{TLL}(\mathcal{D}^\star, \hat{\Pi}) \approx -\mathrm{KL}\left(\mathcal{P}(y^\star) \,\|\, \hat{\Pi}(y^\star|\mathcal{D})\right) + \text{constant}.$$

To illustrate this notion of closeness, or equivalently discrepancy, in Figure 5, we draw a pink line between $\mathcal{P}(y^\star)$ and $\hat{\Pi}(y^\star|\mathcal{D})$. We observe that the TLL importantly does *not* approximate (even up to a constant) the analogous discrepancy from the approximate posterior predictive $\hat{\Pi}(y^\star|\mathcal{D})$ to the exact posterior predictive $\Pi(y^\star|\mathcal{D})$ (blue line in the upper surface); that is, it does not capture how close the posterior predictive approximation is to the exact posterior predictive. The TLL likewise does not approximate (even up to a constant) the corresponding discrepancy from the approximate posterior $\hat{\Pi}(\theta|\mathcal{D})$ to the exact posterior

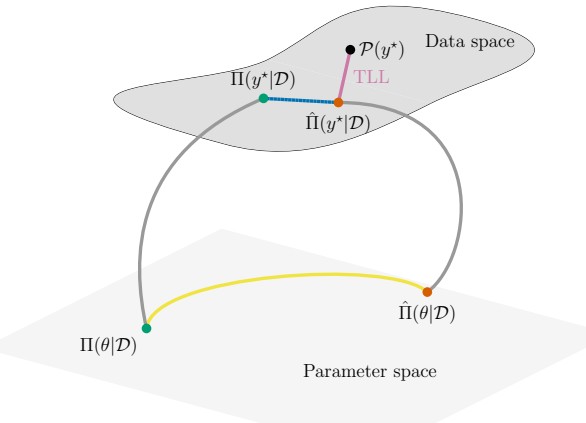

Figure 5: Cartoon illustration highlighting the difference between three different discrepancies explored in Section 3.4. The surfaces are spaces of distributions over a latent parameter (lower surface) or an observable data point $y^\star$ (upper surface). The pink line indicates that $\text{TLL}(\mathcal{D}^\star; \hat{\Pi})$ estimates a discrepancy between the approximate posterior predictive $\hat{\Pi}(y^\star|\mathcal{D})$ (upper surface, lower right, red dot) and the true data-generating distribution $\mathcal{P}(y^\star)$ (upper surface, upper right, black dot). The blue line represents a different discrepancy between the exact posterior predictive (upper surface, left, green dot) and the approximate posterior predictive (upper surface, lower right, red dot). The yellow line represents another different discrepancy between the exact posterior (lower surface, left, green dot) and the approximate posterior (lower surface, right, red dot). Gray lines connect distributions over parameters with their corresponding predictive distributions.

$\Pi(\theta|\mathcal{D})$ (yellow line in the lower surface); that is, it does not capture how close the posterior approximation is to the exact posterior.

The pink and blue lines would (nearly) align if the posterior predictive were very close to the true data-generating distribution. For a misspecified model, the posterior predictive need not be close to the true data-generating distribution. For a well-specified model, the posterior predictive and true data-generating distribution may still be far for a finite dataset. On that view, as suggested by an anonymous referee, we might expect the observed phenomenon to disappear asymptotically in the well-specified setting if sufficient regularity conditions hold. The argument, essentially, is that (i) the actual posterior $\Pi(\theta|\mathcal{D})$ converges to a point-mass at the true data generating parameter; this convergence implies (ii) that the actual posterior predictive $\Pi(y^\star|\mathcal{D})$ converges to the true data distribution $\mathcal{P}(y^\star)$, from which it follows that for large enough training datasets (iii) $\text{KL}\left(\Pi(y^\star|\mathcal{D}) \parallel \hat{\Pi}(y^\star|\mathcal{D})\right) \approx \text{KL}\left(\mathcal{P}(y^\star) \parallel \hat{\Pi}(y^\star|\mathcal{D})\right)$. However, we emphasize first that essentially every real data analysis is misspecified. And second, if a practitioner is in a setting where they are confident there is no uncertainty in the unknown parameter value, there may be little reason to take a Bayesian approach or go to the sometimes-considerable computational burden of approximating the Bayesian posterior.

## 4 Claim: higher test log-likelihood corresponds to lower predictive error

As noted in Sections 2.1 and 3.4, TLL estimates how close a predictive distribution is from the true data-generating process in a specific KL sense. On that view and analogous to Section 3, we would not expect conclusions made by TLL to match conclusions made by comparing other predictive losses. Rather than

focus on more esoteric losses in our experiments, we note that TLL and RMSE are often reported as default measures of model fit quality in papers. If conclusions made between TLL and RMSE do not always agree (as we expect and reinforce experimentally next), we should not expect TLL to always reflect performance according to other predictive losses beyond RMSE. If the TLL is of fundamental interest, this observation is of little consequence; if TLL is a convenient stand-in for a potential future loss of interest, this observation may be meaningful.

**Misspecified Gaussian process regression.** We next construct two models $\Pi$ and $\tilde{\Pi}$ such that $\text{TLL}(\mathcal{D}^\star; \Pi) < \text{TLL}(\mathcal{D}^\star; \tilde{\Pi})$ but $\tilde{\Pi}$ yields larger predictive RMSE. Suppose we observe $\mathcal{D}_{100} = \{(x_n, y_n)\}_{n=1}^{100}$ from the following data generating process:

$$x_n \sim \mathcal{U}(-5, +5) \quad y_n | x_n \sim \mathcal{N}(\sin(2x_n), 0.1). \tag{10}$$

Further suppose we model this data using a zero-mean Gaussian process (GP) with Gaussian noise,

$$f \sim \text{GP}(\mathbf{0}, k(x, x')), \quad y_n | f_n \sim \mathcal{N}(f_n, \sigma^2), \tag{11}$$

where $f_n$ is shorthand for $f(x_n)$. First consider the case where we employ a periodic kernel,[2] constrain the noise nugget $\sigma^2$ to 1.6, and fit all other hyper-parameters by maximizing the marginal likelihood. The resulting fit is shown in Figure 6 (A). Next, consider an alternate model where we use a squared-exponential kernel and fit all hyper-parameters including the noise nugget via maximum marginal likelihood. The resulting fit is displayed in Figure 6 (B). The squared exponential model fails to recover the predictive mean and reverts back to the prior mean (RMSE = 0.737, 95% confidence interval $[0.729, 0.745]$), while the periodic model recovers the predictive mean accurately, as measured by RMSE = 0.355 (95% confidence interval $[0.351, 0.360]$). Despite the poor mean estimate provided by the squared exponential model, it scores a substantially higher TLL.

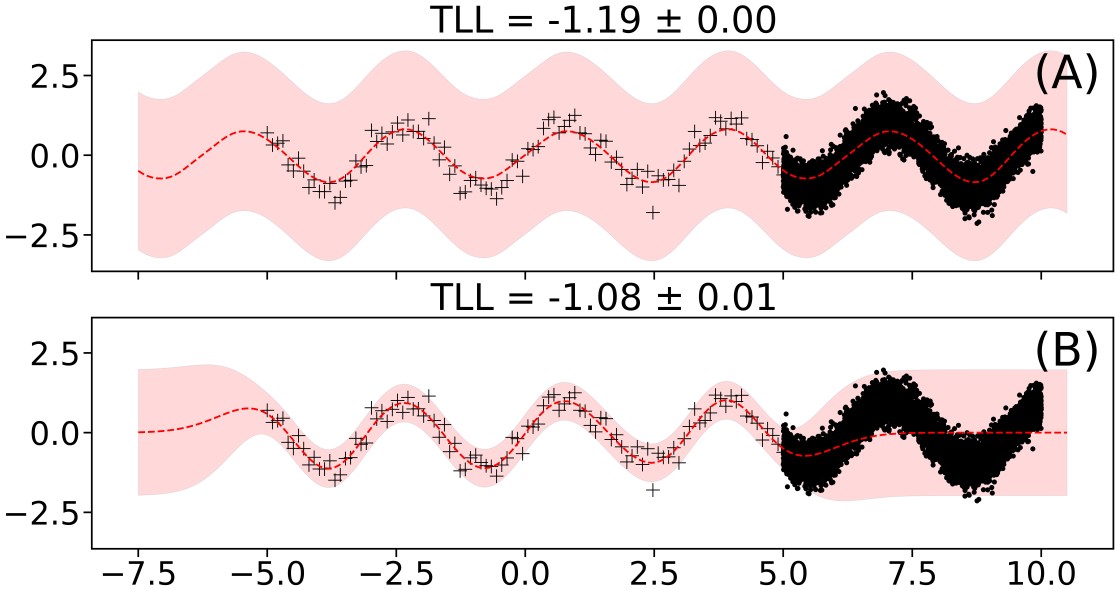

Figure 6: The plots display two Gaussian processes trained on the same set of data (represented by black plus symbols). The dashed red line shows the mean of the posterior Gaussian process, while the red highlighted region represents the 95% predictive interval. The subplot titles display the TLL ($\pm 2$ standard error) attained by each Gaussian process. Although the Gaussian process in panel (A) achieves a better mean fit compared to panel (B), it has a worse TLL when evaluated on $10^4$ test instances (represented by black dots).

---

[2]PERIODICMATERN32 in `https://github.com/SheffieldML/GPy`

In this example, we see that, even with an accurate point estimate, TLL can be reduced by, for instance, inflating the predictive uncertainty. And this discrepancy between TLL and RMSE is not necessarily removed by optimizing the parameters of a model.

**Misspecified linear regression.** Our next example illustrates that even when all parameters in a model are fit with maximum likelihood, a comparison based on TLL may still disagree with a comparison based on RMSE. It also illustrates that the discrepancy between TLL and RMSE can arise even in very simple and low-dimensional models and even when the training dataset is very large.

Specifically, suppose that we observe $\mathcal{D} = \{(x_n, y_n)\}_{n=1}^{100,000}$ generated according to

$$x_n \sim \mathcal{U}(0, 25), \quad y_n | x_n \sim \text{Laplace}(x_n, 1/\sqrt{2}), \tag{12}$$

which we model using one of the following misspecified conditional linear models:

$$\begin{aligned} \Pi : y_n | x_n &\sim \mathcal{N}(\theta x_n, \sigma^2) \\ &\text{or} \\ \tilde{\Pi} : y_n | x_n &\sim \text{Laplace}(0.45 + \theta x_n, \lambda). \end{aligned} \tag{13}$$

Both $\Pi$ and $\tilde{\Pi}$ depend on two unknown parameters. $\Pi$ depends on a slope $\theta$ and a residual variance $\sigma^2$ and $\tilde{\Pi}$ depends on a slope $\theta$ and a residual scale $\lambda$. The kind of misspecification is different across models; while $\Pi$ has the correct mean specification but incorrect noise specification, $\tilde{\Pi}$ has incorrect mean specification but correct noise specification.

We computed the maximum likelihood estimates (MLEs) $(\hat{\theta}_\Pi, \hat{\sigma}_\Pi)$ and $(\hat{\theta}_{\tilde{\Pi}}, \hat{\lambda}_{\tilde{\Pi}})$ for both models. The two fitted models induce the following predictive distributions of $y^\star | x^\star$:

$$\begin{aligned} \Pi(y^\star | x^\star, \mathcal{D}) : y^\star | x^\star &\sim \mathcal{N}(\hat{\theta}_\Pi x^\star, \hat{\sigma}_\Pi^2) \\ &\text{and} \\ \tilde{\Pi}(y^\star | x^\star, \mathcal{D}) : y^\star | x^\star &\sim \text{Laplace}(0.45 + \hat{\theta}_{\tilde{\Pi}} x^\star, \hat{\lambda}_{\tilde{\Pi}}). \end{aligned} \tag{14}$$

The means of these predictive distributions are natural point estimates of the output $y^\star$ at input $x^\star$.

Using a test set of size $N^\star = 395,000$, we observed $\text{TLL}(\mathcal{D}^\star; \Pi) = -1.420 < -1.389 = \text{TLL}(\mathcal{D}^\star; \tilde{\Pi})$. The standard error of either TLL estimate is only 0.002. Hence, based on sample mean and standard error, we conclude that $\tilde{\Pi}$ has better elpd than $\Pi$. These values suggest that on average over inputs $x^\star$, $\tilde{\Pi}(y^\star | x^\star, \mathcal{D})$ is closer to $\mathcal{P}(y^\star | x^\star)$ than $\Pi(y^\star | x^\star, \mathcal{D})$ in a KL sense. However, using the same test set, we found that $\Pi$ yielded more accurate point forecasts, as measured by root mean square error (RMSE):

$$\left( \frac{1}{N^\star} \sum_{n=1}^{N^\star} (y_n^\star - \hat{\theta}_\Pi x_n^\star)^2 \right)^{1/2} = 1.000 < 1.025 = \left( \frac{1}{N^\star} \sum_{n=1}^{N^\star} (y_n^\star - 0.45 - \hat{\theta}_{\tilde{\Pi}} x_n^\star)^2 \right)^{1/2}. \tag{15}$$

In addition, the 95% confidence intervals for the RMSE do not overlap: the interval for $\Pi$'s RMSE is $[0.997, 1.005]$ and that for $\tilde{\Pi}$'s RMSE is $[1.022, 1.029]$. The comparison of RMSEs suggests that on average over inputs $x^\star$, the predictive mean of $\Pi(y^\star | x^\star, \mathcal{D})$ is closer to the mean of $\mathcal{P}(y^\star | x^\star)$ than the predictive mean of $\tilde{\Pi}(y^\star | x^\star, \mathcal{D})$. In other words, the model with larger TLL – whose predictive distribution is ostensibly closer to $\mathcal{P}$ – makes worse point predictions than the model with smaller TLL.

## 5 Discussion

Our paper is neither a blanket indictment nor recommendation of test log-likelihood. Rather, we hope to encourage researchers to explicitly state and commit to a particular data-analysis goal – and recognize that different methods may perform better under different goals. For instance, when the stated goal is to approximate (summary statistics of) a Bayesian posterior, we argue that it is inappropriate to rely on test log-likelihood to compare different approximation methods. We have produced examples where a model

can provide a better test log-likelihood but yield a (much) poorer approximation to the Bayesian posterior – in particular, leading to fundamentally different inferences and decisions. We have described why this phenomenon occurs: test log-likelihood tracks closeness of approximate posterior predictive distributions to the data-generating process and not to the posterior (or posterior predictive) distribution. At the same time, we recognize that evaluating posterior approximation quality is a fundamentally difficult problem and will generally necessitate the use of a proxy. It may be useful to consider multiple of the available options; a full accounting is beyond the scope of this paper, but they include using conjugate models where exact posterior summary statistics are available; comparing to established MCMC methods on models where a sufficiently large compute budget might be expected to yield a reliable approximation; simulation-based calibration (Talts et al., 2018); sample-quality diagnostics (Gorham & Mackey, 2015; Chwialkowski et al., 2016; Liu et al., 2016); and a host of visual diagnostics (Gabry et al., 2019). A careful investigation to understand how a particular method struggles or succeeds may be especially illuminating.

On the other hand, in many data analyses, the goal is to make accurate predictions about future observables or identify whether a treatment will help people who receive it. In these cases and many others, using a Bayesian approach is just one possible means to an end. And many of the arguments for using the exact Bayesian posterior in decision making assume correct model specification, which we cannot rely upon in practice. In predictive settings in particular, test log-likelihood may provide a compelling way to assess performance. In addition to being essentially the only strictly proper local scoring rule (Bernardo & Smith, 2000, Proposition 3.13), TLL is sometimes advertised as a "non-informative" choice of loss function (Robert, 1996). Importantly, however, non-informative does not mean all-encompassing: as our examples in Section 4 show, test log-likelihood does not necessarily track with other notions of predictive loss. As we discuss in Section 2.1, test log-likelihood quantifies a predictive discrepancy only in a particular Kullback–Leibler sense. It is important to note, however, that just because two distributions are close in KL, their means and variances need not be close; in fact, Propositions 3.1 & 3.2 of Huggins et al. (2020) show that the means and variances of distributions that are close in KL can be arbitrarily far apart. So even in settings where prediction is of interest, we recommend users clearly specify their analytic goals and use evaluation metrics tailored to those goals. If there is a quantity of particular interest in the data-generating process, such as a moment or a quantile, a good choice of evaluation metric may be an appropriate scoring rule. Namely, one might choose a scoring rule whose associated divergence function is known to quantify the distance between the forecast's quantity of interest and that of the data-generating process. For instance, when comparing the quality of mean estimates, one option is using the squared-error scoring rule, whose divergence function is the integrated squared difference between the forecast's mean estimate and the mean of the data-generating process. Another option is the Dawid–Sebastiani score (Dawid & Sebastiani, 1999), which prioritizes accurately estimating predictive means and variances. See Gneiting & Raftery (2007) for a list of commonly used scoring rules and their associated divergences.

### Acknowledgments

We are grateful to Will Stephenson for helping us find examples of discrepancies between posterior approximation quality and TLL.

This work was supported in part by the MIT-IBM Watson AI Lab, an NSF Career Award, an ONR Early Career Grant, the DARPA I2O LwLL program, an ARPA-E project with program director David Tew, and the Wisconsin Alumni Research Foundation.

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

## A  Variational Approximations

In Section 3 we formed isotropic Gaussian approximations to the exact posterior. In our illustrative examples, the exact posterior itself is a Gaussian distribution, $\mathcal{N}(\mu, \Sigma)$. In Sections 3.1 and 3.3 we use variational approximations that share the same mean as the exact posterior and are isotropic, $\mathcal{N}(\mu, \rho\mathrm{I})$, where I is a two-dimensional identity matrix and $\rho > 0$ is a scalar. In this family of distributions, the optimal variational approximation is $\mathcal{N}(\mu, \rho^*\mathrm{I})$, where,

$$
\begin{aligned}
\rho^* &= \underset{\rho \in \mathbb{R}_+}{\operatorname{argmin}} \operatorname{KL}\left(\mathcal{N}(\mu, \rho\mathrm{I}) \,\|\, \mathcal{N}(\mu, \Sigma)\right), \\
&= \frac{2}{\operatorname{tr}(\Sigma^{-1})}.
\end{aligned}
\tag{16}
$$

The result follows from setting the gradient $\nabla_\rho \operatorname{KL}\left(\mathcal{N}(\mu, \rho\mathrm{I}) \,\|\, \mathcal{N}(\mu, \Sigma)\right))$ to zero and rearranging terms,

$$
\begin{aligned}
\nabla_\rho \operatorname{KL}\left(\mathcal{N}(\mu, \rho\mathrm{I}) \,\|\, \mathcal{N}(\mu, \Sigma)\right)) = 0, &\implies \nabla_\rho \frac{\operatorname{tr}(\rho\Sigma^{-1})}{2} - \nabla_\rho \ln \rho = 0, \\
\implies \frac{1}{\rho} = \frac{\operatorname{tr}(\Sigma^{-1})}{2}, &\implies \rho = \frac{2}{\operatorname{tr}(\Sigma^{-1})}.
\end{aligned}
\tag{17}
$$

Note that $\rho^*$ is guaranteed to be positive since $\Sigma^{-1}$ is positive definite and thus $\operatorname{tr}(\Sigma^{-1}) > 0$. This optimal variational approximation, $\mathcal{N}(\mu, \rho^*\mathrm{I})$ is used in Panel (B) of Figure 1, Figure 2, and Figure 4. The other panels use $\mathcal{N}(\mu, \lambda\rho^*\mathrm{I})$, with $\lambda \in [1, 5, 10, 15, 30]$ for Figure 1 and Figure 2. For Figure 4 (Left), $\lambda$ takes values in $[4, 5, 7, 9]$, and in $[1, 2, 3, 4, 5, 6, 7, 8, 9, 10, 11]$ for Figure 4 (Right).

## B  Experimental details and additional experiments

### B.1  Confidence Intervals

**An additional note on confidence intervals for TLL.** Suppose we are comparing two models $\Pi$ and $\tilde{\Pi}$. Although $\operatorname{TLL}(\mathcal{D}^\star; \Pi)$ (respectively, $\hat{\sigma}_{\operatorname{TLL}}(\Pi)$) will generally be correlated with $\operatorname{TLL}(\mathcal{D}^\star; \tilde{\Pi})$ (respectively, $\hat{\sigma}_{\operatorname{TLL}}(\tilde{\Pi})$), we do not expect a more careful treatment of that correlation to change our substantive conclusions.

**Confidence intervals for RMSE.** To compute the RMSE confidence interval, we first compute the mean of the squared errors (MSE, $m$) and its associated standard error of the mean ($s$). Since we have a large number of data points and the MSE takes the form of a mean, we assume the sampling distribution of the MSE is well-approximated by a normal distribution. We use $[m - 2s, m + 2s]$ as the 95% confidence interval for the MSE. We use $[\sqrt{m - 2s}, \sqrt{m + 2s}]$ as the 95% confidence interval for the RMSE. Note that the resulting RMSE confidence interval will generally not be symmetric.

### B.2  Additional TLL in the wild experiments

**SWAG with higher learning rates.**  In Figure 7 we continue the experiment described in Section 3.2 but using higher learning rates of 12, 15, and 20. Despite moving further from the exact posterior the test log-likelihood remains higher than those achieved by SWAG approximations with lower learning rates (panels (B) through (E) of Figure 3).

**Mean field variational inference.**  Next, we reproduce the experimental setup described in Section 3.2, but instead of using SWAG to approximate the posterior, we use mean field variational inference and examine the relationship between TLL and posterior approximation quality under different re-scalings of the marginal variance of the optimal variational approximation. Figure 8 shows the posterior mean and the 95% predictive interval of the misspecified regression line $\theta^\top \phi$ from (A) the Bayesian posterior; (B) the mean field variational approximation restricted to isotropic Gaussians; and (C)–(F) several re-scaled variational approximations. In each plot, we overlaid the observed data $\mathcal{D}_{500}$, the true data generating function in dashed black, and also report the 2-Wasserstein distance between the true posterior and each approximation and the TLL

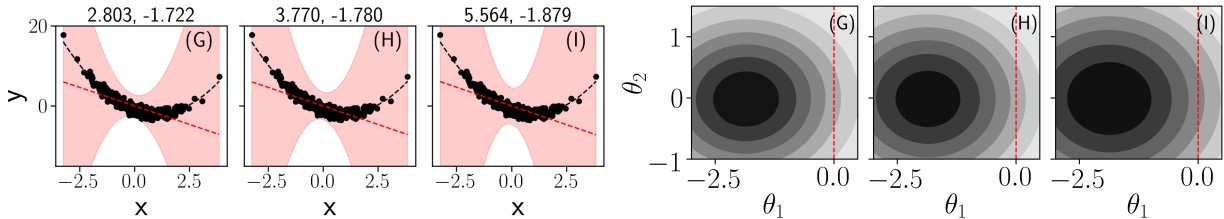

Figure 7: *(Left)*. Predictive distributions under the SWAG posterior with SWAG learning rate of (G) 12, (H) 15, (I) 20. The two numbers in the title of each plot are the 2-Wasserstein distance to the exact posterior and test log-likelihood computed on $10^4$ test set observations. Two standard errors in the test log-likelihood estimates are (G) 0.01, (H) 0.009, (I) 0.08. *(Right)*. Contours of the SWAG approximations with different learning rates. The line $\theta_1 = 0$ is highlighted in red.

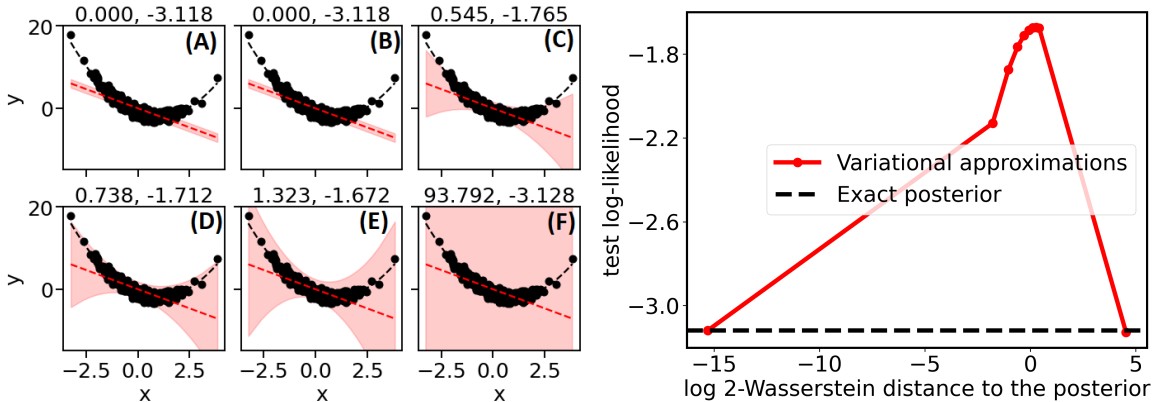

Figure 8: *(Left)*. Predictive distributions under the Bayesian posterior and mean field variational approximations. The two numbers in the title of each plot are the 2-Wasserstein distance to the true posterior and test log-likelihoods computed on $10^4$ test set observations. Two standard errors in the test log-likelihood estimates are (A) 0.16, (B) 0.16, (C) 0.03, (D) 0.02, (E) 0.02, (F) 0.01. *(Right)*. The relationship between distance to posterior and test log-predictive density. Observe the log scale of the horizontal axis and the non-monotonic relationship between test log-predictive density and 2-Wasserstein distance to the Bayesian posterior.

averaged over $N^* = 10^4$ test data points drawn from Equation (4) Like in our previous example, the mean field approximation (panel (B) of Figure 8) is very close to the exact posterior. Further, as we scale up the marginal variance of the approximate posteriors, the posterior predictive distributions cover more data, yielding higher TLL, while simultaneously moving away from the exact posterior over the model parameters in a 2-Wasserstein sense. Interestingly, when the approximation is diffuse enough, TLL decreases, again highlighting its non-monotonic relationship with posterior approximation quality. In this example of a misspecified model, the non-monotonic relationship between TLL and 2-Wasserstein distance means that TLL is, at best, a poor proxy of posterior approximation quality.

### B.3 The highest TLL does not match the best estimate of a posterior summary statistic or the lowest KL

We first reproduce the experimental setup that produced Figure 4, but now in Figure 9, we plot TLL against the error in estimating the posterior standard deviation. In particular, the horizontal axis shows the absolute value of the difference between (a) the marginal standard deviation of the parameters of interest under the approximation and (b) the marginal standard deviation under the exact posterior. As in the right panel of Figure 4, we observe that the highest (best) TLL does not correspond to the lowest (best) error in estimating the posterior standard deviation.

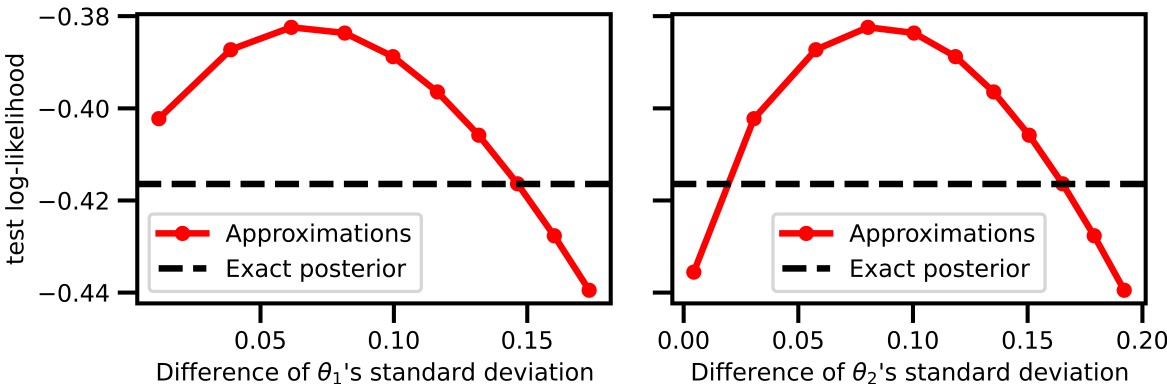

Figure 9: The non-monotonic relationship between difference in marginal standard deviations and TLL in a well-specified case. *(Left)* The horizontal axis reports the absolute difference in the standard deviation of the weight $\theta_1$ between an approximation and the posterior. *(Right)* The horizontal axis reports the absolute difference in the standard deviation of the bias $\theta_2$ between an approximation and the posterior.

To create Figure 10, we reproduce the experimental setup from Figure 8. Relative to the right panel of Figure 8, we change only what is plotted on the horizontal axis; for Figure 10, we plot the log of the absolute value of the difference between marginal standard deviations. Again, we see that the highest (best) TLL does not correspond to the lowest (best) error in estimating the posterior standard deviation.

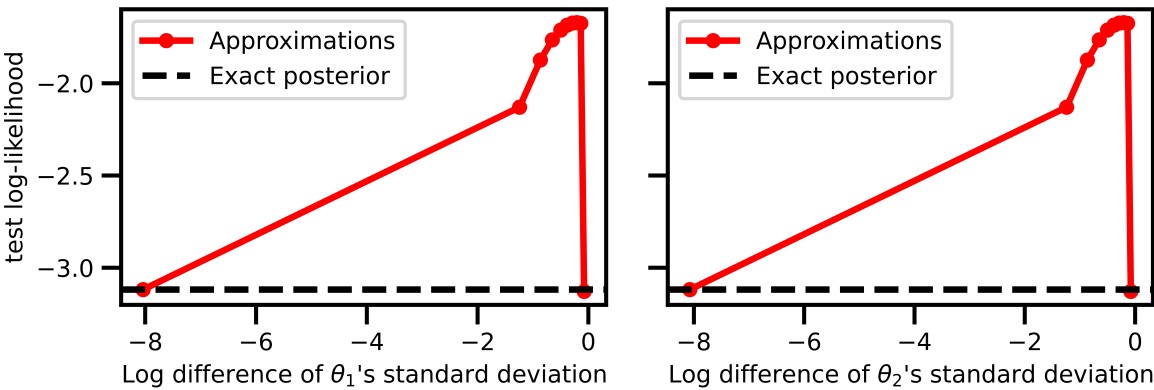

Figure 10: The non-monotonic relationship between difference in marginal standard deviations and TLL in a misspecified case. The meaning of horizontal axis is similar to that of Figure 9.

Finally, Figure 11 reproduces analyses from the main text but uses KL divergence instead of 2-Wasserstein distance to measure posterior approximation quality. In particular, the left panel of Figure 11 recreates the right panel of Figure 1; as in Figure 1, we see that the highest (best) TLL does not correspond to the lowest (best) divergence value. Likewise, the right panel of Figure 11 recreates the right panel of Figure 4; as in Figure 4, we see the highest TLL again does not correspond to the lowest divergence value.

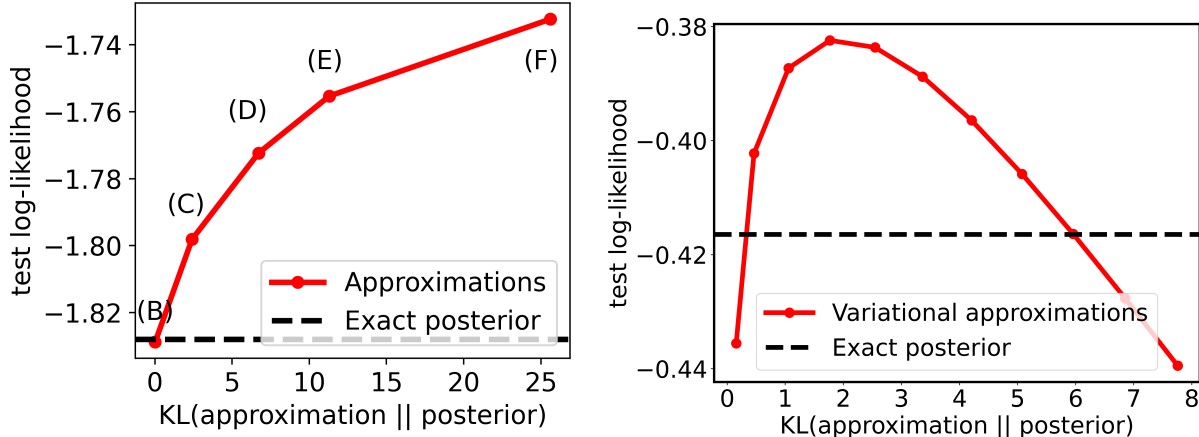

Figure 11: The smallest KL divergence does not correspond to the largest TLL, in a misspecified case (*left*) and a well-specified case (*right*). The left panel reproduces the experimental results presented in the right panel of Figure 1, but uses the reverse KL divergence to measure discrepancy with the exact posterior instead of the 2-Wasserstein distance. The right panel reproduces the results in the right panel of Figure 4.

# C   Alternative model comparison metrics

In this work, we focused on one model comparison metric, test log-likelihood. However, there are many other model comparison metrics. In what follows, we consider marginal likelihood and the joint log predictive density as potential model comparison metrics.

## C.1   Marginal likelihood

Instead of choosing the model with higher test log-likelihood, one might instead choose the model with higher marginal likelihood, where the marginal likelihood for model $\Pi$ is $\pi(\mathcal{D}) = \int \pi(\mathcal{D}|\theta)\pi(\theta)d\theta$. That is, given models $\Pi$ and $\tilde{\Pi}$, one might choose $\Pi$ whenever $\pi(\mathcal{D}) > \tilde{\pi}(\mathcal{D})$;

As an anonymous referee pointed out, the marginal likelihood is a well-established Bayesian model selection criterion; see, e.g., pg. 348 of MacKay (2003). However, the marginal likelihood criterion has several well-known limitations. First, marginal likelihood comparisons can be unreliable when the models being compared are misspecified; see §2 and references within Huggins & Miller (2023), and see also Berk (1966). Beyond this concern, marginal likelihood quantifies only how well a model fits the available training data. It is otherwise silent or "peripherally related" (Lofti et al., 2022) to the predictive quality of the model. And so, just because a model $\Pi$ yields a higher marginal likelihood than $\tilde{\Pi}$, it does not necessarily follow that $\Pi$ produces better predictions of future data than $\tilde{\Pi}$ – even when the training and testing data are i.i.d. from the same distribution.

These limitations notwithstanding, one might still try to use marginal likelihood to assess posterior approximation. Though we have not explicitly constructed examples, we anticipate similar phenomena as reported above; namely, we expect it is possible for one posterior approximation to achieve higher marginal likelihood than another while providing a worse approximation to the exact posterior. And we would expect these phenomena to occur for reasons analogous to the behavior we saw for TLL; what marginal likelihood measures is not directly related to posterior approximation quality.

### C.2   The logarithm of the joint predictive density

Lofti et al. (2022) consider the logarithm of the joint predictive density, $\log \pi(y_1^\star, \ldots, y_{N^\star}^\star | \mathcal{D})$, as an alternative metric for assessment of predictive quality. Analogous to Equation (1),

$$\pi(y_1^\star, \ldots, y_{N^\star}^\star | \mathcal{D}) = \int \pi(y_1^\star, \ldots, y_{N^\star}^\star | \theta) \pi(\theta | \mathcal{D}) d\theta. \tag{18}$$

If the training data are strictly independent from the test data under the specified model $\Pi$ (not just conditionally independent given a latent parameter), then the log joint predictive density will equal the TLL – but such a case would generally be uninteresting in Bayesian modeling. In general, the log joint predictive density need not equal the TLL.

We have already seen that the TLL is a Monte Carlo estimate of the elpd; that is, the TLL is an estimate of "how well is a model *expected* to predict a single held-out observation, *on average*?" The number of samples in that estimate (i.e., the number of test data points) will dictate the Monte Carlo noise of that estimate. There are analogously (at least) two perspectives on the log joint predictive density. One is that the log joint predictive measures "how well does a model predict a specific collection of held-out data?" A second is that the log joint predictive is an estimate (from just a single noisy Monte Carlo draw and therefore very high in variance) of "how well is a model expected to predict the next $N^\star$ held-out observations, on average?" The TLL and log joint predictive can be seen as two extremes of a spectrum, as we describe next. Suppose one could divide a test set of size $N^\star$ evenly into $M$ mini-batches of test data. Then one could make a Monte Carlo estimate with $M$ samples of "how well is a model expected to predict the next $N^\star/M$ held-out observations, on average?" As $M$ increases, the Monte Carlo noise decreases and the size of the held-out set of interest decreases.

Depending on the application, any of these questions may be of interest to a practitioner. While the object of study in this paper has been the TLL, the arguments in Section 3.4 suggest that the log joint predictive density (or any other point on the spectrum above) would exhibit a similar issue to the one described in this paper. After all, just like TLL, log joint predictive density does not directly track discrepancies between distributions of latent model parameters.

