# OpenReview forum: "Are you using test log-likelihood correctly?"
_TMLR — Accepted by TMLR_

### Review · Reviewer_bq8h · 2023-10-16

**Summary Of Contributions:**

The authors raise awareness of issues in using test log-likelihood as metric for evaluating inference techniques, especially from the perspective of using it for selecting which posterior approximations to use. They point out that a worse approximation can result in better test log-likelihood, and that a solution with better test log-likelihood can still have worse predictive accuracy in terms of another metric.

The main message is to warn practitioners about these aspects, in form of concrete examples that make it clear how test log-likelihood for even extremely poor posterior approximations can be the best one, especially for misspecified models.

**Audience:**

Yes

**Broader Impact Concerns:**

None. The paper makes important remarks that should help the community be more transparent in evaluating scientific contributions and hence has only positive ethical implications.

**Claims And Evidence:**

No

**Requested Changes:**

There is clear need for a paper that discusses the topics, since it indeed has become common to access various posterior approximation techniques in terms of TLL in large-scale problems where direct evaluation of the approximation quality is not possible. In other words, I agree that various authors implicitly rely on the first claim as basis for their empirical evaluation. Even though the observations made here are not strictly new and most researchers working with the methods are likely aware that TTL is merely a proxy and not a direct measure of the approximation quality, it is highly useful to have a paper that can be conveniently cited e.g. to justify why measuring directly the approximation quality instead of "the standard TLL that others use".

However, I am not so sure whether the second claim is really being used. I have always thought that people report TLL and RMSE separately as the most common metrics exactly because they are well aware you cannot make conclusions across the two metrics. The experimental results in Section 4 are valuable in explicating the discrepancy, but I do not think there is convincing case of calling that particular misuse a common scenario. In light of this, I would like to see the statements in Introduction to be reformatted -- I do not think the paper would be any weaker even if the second claim was characterised from the perspective of "we additionally show how ... as a reminder that different test accuracy metrics do not necessarily align", instead of claiming that particular misunderstanding to be common. As a further evidence of Claim 1 being more believable: In Section 3 (2nd paragraph) you give a long list of references to papers that rely on this implicit claim, whereas in Section 4 you do not have any examples like this. Either you need to find those references, or should fine-tune the narrative.

I like the synthetic examples as they explain very clearly how the effects are seen throughout very broad range of approximation widths; e.g. Figures 3 and 4 shows that we can indeed have approximation widths that are off by an order of magnitude and still get nice TLL. This is, in my opinion, better approach than simply showing differences in ordering for some real approximations computed with different methods (that would have fairly similar approximations and TLLs).

Even though I see value in the paper, I was a bit disappointed on the depth of the discussion and lack of concrete recommendations. I understand that you are not willing to directly say "don't use TLL to compare approximations", but the paper would still be stronger if there was an actionable recommendation for the authors who are working on approximate inference methods in context of large problems where explicit comparison against true posterior is not feasible. For instance, you could recommend some sort of template sentences to use when explaining limitations of TLL, as well as alternative metrics people should routinely report. What would be your best recommendation for a metric we could use to compare e.g. Bayesian NN posteriors, given that TLL is not a valid measure? Or would you still say that in absence of better measures we should keep on reporting TLL but explain more transparently why it could lead to (potentially severe) mis-ranking of alternative approximations. It would be very useful to have a clear statement on this.


**Other remarks:**
- A bit more discussion on epld and Dawid-Sebastiani scores would be useful (e.g. definition of the latter). Section 2.1 now kind of suggests epld would be a more useful quantity to estimate directly and it would be better to discuss why people are not doing this but are using TTL instead. If we were (able to) estimate epld directly, you would get rid of the implicit assumption of TTL being a close approximation to epld at the end of Section 2.1, so it would be useful to explain for the readers the challenges in that.

- You occasionally talk about "model comparison" (e.g. end of Section 2.1, beginning of Section 3.1), in places where I believe "approximation comparison" would be more appropriate. I think the misuse happens more commonly in cases where people compare different inference methods, not different models. This is what most of the examples in the beginning of Section 3 are doing.

- Formatting: You heavily use footnotes to explain rather complex aspects. At least footnotes 1 and 7 are quite detailed and important parts, so I would integrate them into the main body.

- Would be good to discuss the choice of 2-Wasserstein as the sole measure of approximation quality. At least give a definition and explain why you used it, since it is not really (yet?) a standard metric in the field. Would your results look the same if you used e.g. (symmetric) KL-divergence to measure the distance to the true posterior? Is 2-Wasserstein in some sense ideal measure here, or does it have limitations/weaknesses as well?

**Strengths And Weaknesses:**

**Strengths**
1. The message is very important. Test log-likelihood is indeed often used as primary or even sole measure of approximation quality, and it is not sufficient.
2. Even though the basic observations are largely known within the community, we do not have clear references that could be used to justify the choices of the metrics. This paper could serve as one.
3. The examples where approximation width is artificially changed are innovative and clear.

**Weaknesses**
1. The paper fails to provide very concrete recommendations or suggestions as replacement (or for complementing) TLL as the evaluation metric, limiting the impact.
2. The second claim feels a bit like straw man; I do not believe people misuse test log-likelihood in that sense.

---

> ### Author Response · Authors · 2023-11-05
> **Response to bq8h**
>
> **It is highly useful to have a paper that can be conveniently cited:**
> Thank you for contextualizing our work in this way. Indeed,  while the issues with using TLL as a measure of posterior approximation quality is known to a subsection of the community, the arguments highlighting these problematic issues are scattered through the literature. Our goal with this paper is to both collect these arguments in a single manuscript and to clearly illustrate the issues under  experimental setups that are easy to understand and reproduce.
>
> **Lack of concrete recommendations:**
> We have expanded Section 5 to include a more detailed discussion of alternate recommendations. Broadly, we hope to encourage researchers to explicitly state and commit to a particular data-analysis goal  and recognize that different methods may perform better under different goals. In particular, when the goal is to approximate the moments of a Bayesian posterior, the use of test log-likelihood to compare different approximation methods is inappropriate. While we do not believe simple alternatives exist, we do provide some concrete recommendations for a practitioner in Section 5.
>
> **The second claim feels a bit like straw man:**
> We concede the reviewer’s point that although the TLL is commonly used as a proxy for posterior approximation error, there are far fewer instances in which it is explicitly recommended instead of other loss functions like RMSE. And we are grateful for their suggestion reframing the narrative. We have revised the exposition in Sections 3.4 and 4 along these lines to make the following points:
>
>   * TLL estimates a very specific notion of predictive discrepancy.
>   * On that view, we should not expect conclusions made based on a comparison of TLL to track with conclusions drawn on other comparisons.
>   * Although this expectation may strike some readers as obvious, we nevertheless felt it important to include explicit examples of this phenomenon, for the sake of completeness.
>
> **A bit more discussion on epld and Dawid–Sebastiani scores would be useful:**
> We have substantially revised Section 2 of the manuscript and have moved discussions of alternate scoring rules (including the Dawid–Sebastiani score) to Section 5.
>
> **You occasionally talk about "model comparison" (e.g. end of Section 2.1, beginning of Section 3.1):**
> We have revised the text just before Section 2.1 to clarify this issue.
>
>
> **2-Wasserstein as the sole measure of approximation quality:**
> We have expanded the appendix to highlight that nothing substantive changes if we use alternative discrepancies to the true posterior. We have added an additional figure in the appendix (Figure 11) that is analogous to the plots in Figures 1 and 4 but uses the reverse KL divergence as opposed to the 2-Wasserstein distance. We observe the same behavior between TLL and KL as with TLL and the 2-Wasserstein distance: namely, trends in TLL need not reflect trends in the divergences, and the model with best TLL does not  correspond to the model with best divergence. Moreover, Figures 9 and 10 plot the difference of marginal standard deviations of the parameters of interest between an approximation and the exact posterior vs. TLL and illustrate similar discrepancies. Together, these examples highlight that the trends in TLL need not be meaningfully related to a variety of sensible measures of posterior closeness.
>
> We also hope that our new Section 3.4 will highlight the intuition behind why we should expect the behavior we do, regardless of the choice of divergence.

---

> > ### Comment · Reviewer_bq8h · 2023-11-15
> > **Acknowledging response**
> >
> > I thank the authors for the detailed feedback and acknowledge that I have read both the response and the revised paper. The revision improves the narrative and avoids making unnecessary claims regarding prevalence of misinterpreting the relationship between TLL and MSE.  The new Section 3.4 is helpful in explicating that the true posterior predictive distribution for exact Bayesian inference may not align with the data generating process and the revised Discussion is stronger, more clearly stating that researchers should avoid making conclusions about approximation quality based on TLL and providing some pointers to alternative methods that could be considered.
> >
> > In brief, I do not have any open remarks regarding the paper.

---

### Review · Reviewer_m596 · 2023-10-18

**Summary Of Contributions:**

The paper discusses two claims about Bayesian modelling:

Claim 1: The higher the test log-likelihood, the more accurately an approximate inference algorithm recovers the Bayesian posterior distribution of latent model parameters

Claim 2: The higher the test log-likelihood, the better the predictive performance on held-out data according to other measurements, like root mean squared error.

The way that these claims are addressed is primarily via examples.

I find claim 1 most interesting, as will be discussed below. Secs 3.1 and 3.2 consider mis-specified models, while sec 3.3 considers a well-specified model.  For claim 2, in the era of end-to-end learning, it seems clear that if one wants to optimize RMSE, one should create an objective (e.g.  based on leave-one-out CV) that evaluates/estimates RMSE, and optimize that. There is no particular reason why TLL should be well-correlated with the alternative objective.

The paper lacks experimental details and derivations; these should be provided in appendices, and code also provided.

**Audience:**

Yes

**Claims And Evidence:**

No

**Requested Changes:**

Fundamentally I believe the paper is asking the wrong questions.  It is using an incorrect version of the TLL for the Bayesian case, where it should be using $L(\mathbf{y}^*)$.  The paper needs to be redone to consider $L(\mathbf{y}^*)$ and not the TLL as defined in eq (1).

The examples in secs 3.1 and 3.3 are quite interesting, and could potentially make an interesting paper about some seemingly counter-intutive aspects of Bayesian analysis, where an approximate posterior gives rise to better TLL than the exact posterior. But this will require an entire re-write of the paper. I do not believe that this can be properly completed within the 2 weeks timeframe of TMLR. Thus I believe the correct decision on this paper will be reject, with leave to resubmit.

For detailed comments on sec 3.1, please see above.

The point made in sec 3.3 where an incorrect posterior can give rise to higher TLL is of also interest, but it needs much more exploration to really explain what is going on here. I believe my analysis is very germane to this.

For the reasons given above, I do not find sec 3.2 and sec 4 to be of sufficient interest and recommend that they be cut.

**Strengths And Weaknesses:**

It is worth starting out to say that Bayesian orthodoxy would focus on the marginal likelihood $p(D) = \int p(D|\theta) p(\theta) d\theta$ as a way of comparing models. There are reasons why one might consider the ELPD instead; see e.g. secs 3.8.4-3.8.6 in Murphy PML2 (2023) for discussion on this and related topics.

Certainly TLL makes sense wrt the standard supervised learning setup.  However, in the Bayesian view, we should actually consider $p(\mathbf{y}^*|{\cal D}) = \int p(\theta|{\cal D}) [ \prod_{i=1}^{N^*} p(y^*_i|\theta)] d \theta$, where $\mathbf{y}^*$ is the vector of $N^*$ test cases. I.e. in the Bayesian view the test cases are *conditionally* independent given $\theta$, rather than being fully independent.

The paper asks "Are you using the test log-likelihood correctly?".  Given the above, one answer is no, because one should be looking at $1/N^* \log p( \mathbf{y}^*| {\cal D})$, and not $\sum_{i=1}^{N^*} \log p (y^*_i|{\cal D})$, which is considered here.

Beyond this point, I do not find that the examples argue against using the test log likelihood, especially the correct version $L(\mathbf{y}^*) = 1/N^* \log p(\mathbf{y}^*|{\cal D})$.  To me, what the paper is really about is the initially surprising result that in the examples of sec 3.1 and 3.3, it is the case that using an approximate posterior can give better TLL than using the exact posterior. This can be made sense of more thorough investigation -- see my comments below.

### Example 3.1.

The data (N=100) is generated from a heteroscedastic model, as in eq 2. The mis-specified model is linear regression with unknown offset and slope (theta vector), and a fixed variance of 1.  For this simple setup, it is possible to compute exact Bayesian inference for the theta vector, and also a "mean field" variational approximation. Fig 1 panels C-F show "variational approximations with re-scaled marginal variances", but absolutely no details are given of this rescaling. All these experimental details and derivations (including exact and VI) should be provided in an appendix, and code also provided. Details should also be given (in the appendix) for the computation of the 2-Wasserstein distance (it is a closed form for Gaussians, I believe).

The key observation from this experiment is that the test log likelihood (on a sample of size $N^*=10^4$) is higher for panels C-F, where presumably the posterior variance has been inflated.  So for this mis-specified model, one can obtain higher TLL for worse posterior predictions.

Fig 1 (left) shows why this result is not unexpected -- inflating the variance of the posterior "wobbles" the theta_1 parameter (coeff of x) more, and this translates to error bars that increase with |x|, as in panel F. This mimics (to some extent) the effect of the heteroscedastic variance. But it would be very interesting to see how this works for $L(\mathbf{y}^*)$, as the plots show the marginal erorr bars for one test point, while this quantity requires all $N^*$ datapoints *to be generated by the same $\theta$.*

I would also like to see the marginal likelihood (or ELBO for VI) for each model A-F, and the true test log likelihood $L(\mathbf{y}^*)$.  Also it is readily possible to compute the true log-likelihood under the model of eq 2, this should be given when comparing to the marginal likelihood/ELBOs of the mis-specified models. Please also add a panel to Figure 1 (left) showing the true data generator and error bars -- this will be useful to compare with panel A-F.

### Sec 3.2.

This is a very similar example to sec 3.1, with a mis-specified linear model compared to an exact data generator, here incorporating a quadratic term (eq 4). Similar patterns are observed as in the previous example, with higher TLL for the inexact posteriors.

This example is presented as being "in the wild". However it is again on a toy example (where one can easily see how posterior variance inflation will help explain the unmodelled quadratic component).  The "in the wild" claim presumably relates to the use of SWAG (Maddox et al, 2019) which is claimed to be an off-the-shelf approximate inference algorithm.  However, SWAG essentially fits a (low-rank) Gaussian to the trajectory of a SGD optimization method (with learning rates set in a particular way). The main issue is that there are no guarantees that the method produces a correct estimate of the posterior distribution. Indeed the variance inflation in Fig 3 (right) is not at all unexpected given the higher learning rates used (p 6 para 1).

I believe one could cover the use of SWAG by commenting at the end of sec 3.1 that SWAG with increasing learning rates mimics the effect of variance inflation.

In my view, this example 3.2 thus does not add significantly to that of sec 3.1.

### Sec 3.3. Well-specified models.

In this example, exact posterior analysis is carried out for a small dataset size for a parameter value $\theta_{*} = [-2,-1]$ which is unlikely under the heavily correlated prior of eq 6, as illustrated in Fig 4A. Again here variance inflation can lead to a higher TLL.

This is initially surprising, in that doing exact Bayesian inference in a well-specified model is "doing the right thing". So how is it possible to do better, and that $\Delta(p,q|D) = elpd(p|D) - elpd(q|D)$ is not $\ge 0$?

My analysis explaining this is given below.


For a given datset ${\cal D}$ and testpoint $x_*$ we have

\$p(y_*|x_*, {\cal D}) = \int p(y_*|x_*,\theta) p(\theta|{\cal D}) d\theta,\ $

\$q(y_*|x_*, {\cal D}) = \int p(y_*|x_*,\theta) q(\theta|{\cal D}) d\theta.\ $

Hence the elpd at location $x_*$ for the exact and approximate
methods is given as

$ \mathrm{elpd}(p,x_*|{\cal D}) =  \int \log p(y_*|x_*, {\cal D}) p(y_*|x_*,\theta_*) dy_*$ ,

$ \mathrm{elpd}(q,x_*|{\cal D}) =  \int \log q(y_*|x_*, {\cal D}) p(y_*|x_*,\theta_*) dy_*$ .

Notice that the integration here is with respect to
$p(y_*|x_*,\theta_*)$ and not $p(y_*|x_*,D)$. This is critical for
understanding what is going on. We have the graphical model
$y_* \leftarrow \theta_* \rightarrow {\cal D}$, so that ${\cal D}$ depends on
$\theta_*$, but is not a function of it.

We now integrate the expressions for $\mathrm{elpd}(p,x_*|{\cal D})$ and $\mathrm{elpd}(q,x_*|{\cal D})$
over $p(x_*)$  to obtain

$\mathrm{elpd}(p|{\cal D}) = \int \mathrm{elpd}(p,x_*|{\cal D}) p(x_*)
dx_* $ ,

$\mathrm{elpd}(q|{\cal D}) = \int \mathrm{elpd}(q,x_*|{\cal D}) p(x_*) dx_* $.


Finally we consider the difference of these two quantities:

$\Delta(p,q|{\cal D}) =   \mathrm{elpd}(p|{\cal D}) - \mathrm{elpd}(q|{\cal D}) = \int \int \log \frac{p(y_*|x_*,
    {\cal D})}{q(y_*|x_*, {\cal D})} p(y_*|x_*,\theta_*) p(x_*) dx_* dy_*$ .

If $p(y_*|x_*,\theta_*)$ and $p(y_*|x_*,{\cal D})$ were the same thing, this
would give rise to a KL divergence term averaged over $p(x_*)$, and we would
obtain non-negativity for the elpd difference. In general this does
not hold. However, if $p(\theta|{\cal D})$ converges to $\delta(\theta -
\theta_*)$ for large training sample sizes (as would normally be
expected), then the KL result will hold.

If you use the above analysis in this or a subsequent paper, I request that you make sure to acknowledge the anonymous referee who provided it, and not represent it as your own work.

Given this analysis, it is of interest to know how often we will find $\Delta(p,q|D)$ to be negative, For a fixed $\theta_*$ this can be addressed by repeatedly sampling datasets. It is also important to know how this depends on $\theta_*$; my intuition is that the more probable $\theta_*$'s under the prior will be less likely to give rise to negative differences.

The analysis in section 3.3 is very incomplete, and needs to be expanded to address the issue of *why* the TLL differences occur. It may be good to keep the same test set for all cases, and look at differences in $elpd(p,x_*|D) - elpd(q,x_*|D)$ on a case-by-case basis.


### Sec 4, Claim 2.

Claim 2 seems to me to be much less interesting than Claim 1. Already on page 1 the authors mention some examples highlighting discrepancies between test log-likelihood and other analysis objectives.

The GP example is annoying, in that to make things come out well for the incorrect model (panel B in Figure 5), the authors do two things: (i) sample test data in the interval [5,10], which is disjoint from the training data, and (ii) handicap model A by setting its noise variance too high. But in general in machine learning, if the training and test distributions are different (issue (i)), all bets are off.

The other point here is that if we wish to optimize for (say) RMSE, it is natural to create an objective (e.g.  based on leave-one-out CV) that evaluates/estimates RMSE, and optimize that. For example with GPs it is efficient to compute LOO predictions (see e.g.  Rasmussen and Williams, 2006, sec 5.4.2) and one could optimize this wrt kernel parameters.

### Other points

The title is opaque and inadequate -- it is essential that it at least mentions Bayesian somewhere. The real issues, as discussed above, is that the examples show that approximate Bayesian methods can sometimes give rise to better predictive performance than exact Bayesian methods.

The paper lacks experimental details and derivations; these should be provided in appendices, and code also provided.

In Figs 1 and 4 I find no use for the two numbers above each plot -- these can be obtained from the plots on the RHS. The same could hold for Fig 3 if another plot were made like Fig 1 (right).

Fig 1 caption. The explanation of labels A-F should occur in the caption. A figure+caption must always be self-contained.

---

> ### Author Response · Authors · 2023-11-05
> **Response to m596**
>
> **Alternate criterion for model comparison:**
>
> We thank the reviewer for highlighting an alternate criterion for model comparison.
> While we agree that studying the operating characteristics of such a criterion relative to the test log-likelihood (TLL) would be very interesting, we believe that such a study is beyond the scope of our paper. Rather, our paper examines the widespread practice of using TLL (where -TLL is also commonly called the negative log predictive density, or NLPD) for evaluating the quality of posterior approximations and predictions. We focus on TLL = $\frac{1}{N^{\star}}\sum_{n = 1}^{N^{\star}}{\log \pi(y^{\star}_{n} \vert \mathcal{D})}$ (see Equation 18 in Quiñonero-Candela et al., 2005 or the Wikipedia article for "negative log predictive density") precisely because it is used by virtually every paper that reports a notion of held-out likelihood (e.g., Hoffman et al., 2013; Ranganath et al., 2014; Hernández-Lobato et al., 2016; Liu and Wang, 2016; Shi et al., 2018, Hernández-Lobato and Adams, 2015; Gan et al., 2016; Li et al., 2016; Louizos and Welling, 2016; Sun et al., 2017; Ghosh et al., 2018; Mishkin et al., 2018; Wu et al., 2019; Izmailov et al., 2020; 2021; Ober and Aitchison, 2021).
>
> We believe our paper highlights important distinctions between the TLL, Bayesian posterior approximation, and other notions of predictive quality that are distinct from considerations of how data points are subsetted for testing.
>
> We have updated our manuscript to clarify the provenance and relevance of both TLL and elpd (the expectation of TLL over infinite test data). We have also made the popular usage of the alternative name NLPD more explicit. We have clarified our assumptions in the present manuscript that all data is i.i.d. from a true data-generating distribution. There are many interesting questions that arise in spatiotemporal domains that are also outside the scope of the present paper.
>
>
> **Apparent conflict between our empirics in Section 3.3 and the notion that "doing exact Bayesian inference in a well-specified model is 'doing the right thing'":**
>
> While we appreciate the reviewer’s thoughtful attempt at resolving this conflict, we respectfully reject the premise that "doing exact Bayesian inference is 'doing the right thing'.’’  Allow us to respond to a slightly more precise formulation: that in a well-specified setting in which data are drawn i.i.d. from a fixed parameter value, the exact Bayesian posterior predictive offers the best predictions. If one measures predictive quality using any strictly proper scoring rule, then (1) the best predictions are (by definition) delivered by the actual data generating process (which is unknown) and (2) the exact Bayesian posterior predictive will not exactly coincide with the data generating process.
>
> More formally,  if $\hat{\Pi}(\theta \vert \mathcal{D})$ is an approximation of the actual posterior $\Pi(\theta \vert \mathcal{D})$, then TLL$(\hat{\Pi})$ estimates the negative KL divergence from the approximate predictive to the true data generating process. Importantly it neither measures the KL divergence between the approximate posterior predictive and actual posterior predictive (which is a rather indirect notion of approximation accuracy) nor any discrepancy between $\hat{\Pi}(\theta \vert \mathcal{D})$ and  $\Pi(\theta \vert \mathcal{D})$. Please see the newly added Section 3.4 and Figure 5 in the paper that talk about these issues in greater detail, and also see our response to MA33 about a discussion of these issues in light of the Bernstein–von Mises theorem.
>
> **Experimental results are not unexpected:**
> We are glad that the reviewer does find the experimental results obvious. Our goal with the experiments in Sections 3.1, 3.2, and 3.3 is to provide clear, illustrative examples that make it painfully obvious what test log-likelihood is measuring and how that does not necessarily correlate with better estimates of the moments of the exact Bayesian posterior.
>
>
> **SWAG may not provide correct estimates of the posterior:**
> Indeed, we are counting on the fact that it does not. SWAG with increasing learning rate provides progressively worse estimates of posterior moments, and hence provides incorrect “estimates of the posterior,” and yet increasing learning rate (up to a point) results in higher TLL scores.
>
>
> **Code and derivations:** We have added our derivations to the main text and to the appendix for completeness. We apologize for our oversight in not having uploaded the code earlier; we have now uploaded the code.

---

> ### Comment · Reviewer_m596 · 2023-11-18
> **Response to authors' comments and revised paper**
>
> I have read the reviews, the authors' responses, and the revised paper.
>
> I feel that most of the points that I made in my review have not been addressed
> properly.
>
> 1.  In my review I point out that in a Bayesian analysis, the training (and test) examples are *conditionally* independent given the parameters, not independent, even if the data generator may be assumed iid.  This means that one should naturally consider the marginal likelihood (or ELBO for variational approximation) on the training data, and what I call $L(\mathbf{y}^*)$ for a test set quantity, and not the TLL. This is because the $\mathbf{y}$ or $\mathbf{y}^*$ values are conditionally independent given $\theta$.  In their response the authors point to a lot of papers using TLL -- but just because lots of papers do something does not make it correct.  Indeed MacKay (ITILA book, 2003, p 348) calls the marginal likelihood [there termed the evidence] "a transportable quantity for comparing alternative models".  Reviewer bq8h comments that "I understand that you are not willing to directly say 'don't use TLL to compare approximations', but the paper would be stronger if there was an actionable recommendation for authors ..."  Thus I believe that given that the marginal likelihood (and ELBO for variational inference) is a standard Bayesian quantity used for model comparison, it should be covered in this paper, and not doing so is an unreasonable gap. (I do discuss the marginal likelihood in my original review.)
>
> 2. In my review I highlight the test-set quantity $L(\mathbf{y}^*)$, which is the test analogue of the marginal likelihood. It is worth saying that if we imagine the training and test data as one big dataset, with the first $N$ points being the training set, then $L(\mathbf{y}^*)$ coincides (up to a scaling factor) with conditional log marginal likelihood (CLML) $\log p(D_{> N}|D_{\le N})$ defined in "Bayesian Model Selection, the Marginal Likelihood, and Generalization" by Lofti et al, ICML 2022. The discussion of marginal likelihood and generalization  in that paper is also very relevant to the paper under review, and should be discussed therein.  For the simple linear-Gaussian models used here, it should be possible to evaluate both the marginal likelihood/ELBO, and $L(\mathbf{y}^*)$ analytically.
>
> 3. Although the setup of the paper is concerned with Bayesian inference and its  approximation, the analyses in section 3 concern examples with fixed parameter values. However, in relation to the responses to my comments re sec 3.3. and "doing the right thing", the optimality of Bayesian inference actually relates to average-case analysis (see, e.g.  Average Case epsilon-Complexity in Computer Science: A Bayesian View. Kadane, Joseph B.; Wasilkowski, Grzegorz W., in Bayesian statistics, J. M. Bernardo, ed., pp. 361-374, 1985).  Hence my point about examining how the effect observed in sec 3.3 depends on the true $\theta_*$. It would thus expected that using the exact Bayesian posterior will do better than an inflated version *on average*.
>
> 4. I agree with reviewer bq8h that the second claim is a bit of a straw man. This agrees with my comments that sec 4 should be cut.
>
> 5.  Sec 3.1. I made some comments in my review about how this section could be improved, *explaining* why the TLL is better for the inflated posterior, and how Fig 1 can be improved, e.g. by adding a panel with the true data generator (which would show the heteroscedastic variance graphically). I am disappointed that the authors did not follow this advice, especially wrt the *understanding* of why the TLL improves with increasing posterior variance, up to a point.
>
> 6. TLL and elpd. For the simple linear regression models considered here, it should be possible to do better than the TLL approximation of the elpd (which uses $N^*$ samples). If we consider the expressions for elpd(p|D) and elpd(q|D) given in my review, it may be possible to evaluate these analytically. If not, one should consider numerical quadrature, e.g. Gaussian quadrature.
>
> 7. Other points
>
> - Sec 3.1 please point to Appendix A when you first discuss the isotropic Gaussian approximation, shortly after eq 3. Also it would be helpful to mention the specific variance inflation factors in the main text.
>
> - Figure 1 caption -- explanation of the labels should occur in the caption.
>
> 8. Overall. The paper provides two interesting examples in secs 3.1 and 3.3. However, the analysis of these cases does not take into account standard quantities in Bayesian analysis such as the marginal likelihood/ELBO, and CLML. In my view this is a big gap that needs to be filled before the paper is suitable for publication.

---

> > ### Author Response · Authors · 2023-11-22
> > **Thanks!**
> >
> > We are grateful for the referee’s reply and feedback. We wish to clarify several points.
> >
> > First, the referee suggests the marginal likelihood (MLL) of the data (or evidence) as a standard model comparison criterion. While we agree that MLL is a useful quantity from the perspective of Bayesian hypothesis testing and model selection, we consider it lacking in at least two respects. First, it is well-known that Bayesian model selection (which, essentially, amounts to comparisons of marginal likelihoods) can be unstable/unreliable when all models under consideration are misspecified (see Section 2 of [Huggins & Miller (2020)](https://arxiv.org/pdf/1912.07104.pdf) and references therein; see also [Berk (1966)](https://projecteuclid.org/journals/annals-of-mathematical-statistics/volume-37/issue-1/Limiting-Behavior-of-Posterior-Distributions-when-the-Model-is-Incorrect/10.1214/aoms/1177699597.full). Second, MLL quantifies only how well a model fits the available training data but is silent about how well the model generalizes: just because a model $\Pi$ yields higher MLL than $\tilde{\Pi}$ (i.e. $\pi(\mathcal{D}) > \tilde{\pi}(\mathcal{D})$) does not necessarily mean that $\Pi$ yield better predictions of future data. As we point out in Section 2 and 3.4 of our revised manuscript, TLL, on the other hand, is intimately tied with predictive quality, albeit in a narrow KL sense.
> >
> > Nevertheless, it is not impossible to imagine a practitioner who uses MLL to assess posterior approximation quality. Though we have not explicitly constructed examples, we would anticipate similar phenomena as reported in our present paper. Namely, that it is possible that a posterior approximation $\hat{\Pi}_1$ achieves higher MLL than another posterior approximation $\hat{\Pi}_2$ but yields worse approximation when the goal is to make decisions using the exact posterior mean or variance.
> >
> > The referee then suggests using the joint posterior predictive density (or the CLML of Lofti et al., which is related). We believe it is important to point out that CLML/joint predictive density and TLL provide quantitative answers to two rather different questions. Whereas CLML answers “how well does my model predict this specific set of testing data”, TLL answers “how well can I expect my model to predict a single testing observation, *on average*.” While we believe that both questions have merit, we are unaware of a first-principles argument motivating the use of CLML. One can, however, justify the use of TLL (as a specific predictive criterion, if not a posterior approximation criterion) by appealing to the theory of proper scoring rules. Specifically, TLL is a Monte Carlo approximation of ELPD. ELPD is maximized by the true data generating distribution, and a model achieving larger ELPD yields a predictive that is closer to the truth (in a specific KL sense) than one with smaller ELPD. As far as we can tell, CLML is not related to a proper scoring rule. Is it the case that the true data generating distribution will achieve the highest CLML? And is it the case that a model yielding higher CLML is closer to the true data generating distribution (in any sense)? We believe that (i) it is important to answer these interesting questions before considering how well- or mis-aligned CLML is with posterior approximation quality but that (ii) doing so is well beyond the scope of this paper.
> >
> > The referee introduces a distinction about (conditional) independence. This distinction is immaterial in our setting, since we never assume that the models are correctly specified. From a nonparametric point of view, treating the y*’s as independent is functionally no different than treating them conditionally independent given (the unknown and possibly infinite-dimensional) parameter $\mathcal{P}$). The SLLN and CLT in Section 2 follow from the (conditional) independence of $y^{\star}$ (given $\mathcal{P}$) and not from any dependence assumption encoded in the (certainly mis-specified) model $\Pi(y, \theta)$.
> > We will add a discussion summarizing these points (and citations to the papers mentioned above) to our manuscript.

---

### Review · Reviewer_MA33 · 2023-10-23

**Summary Of Contributions:**

By giving examples with experiments, they refuted the following statement:

* In Section 3, “The higher the test log-likelihood, the more accurately an approximate inference algorithm recovers the Bayesian posterior distribution of latent model parameters.”
* In Section 4, “The higher the test log-likelihood, the better the predictive performance on held-out data according to other measurements, like root mean squared error.”

In the last section, the author suggested that it is important understand that different analysis requires different goals with several suggestions:
* If the goal is approximating posterior, don’t use TTL.
* Change scoring rule depending on goals.

**Audience:**

Yes

**Claims And Evidence:**

Yes

**Requested Changes:**

### Sections 1, and 2

Clearly written

#### Section 3

This section is still clear. However, readers would appreciate more high-level explanations for this phenomenon, not necessarily specific to each instance. For example, in Section 3.3, I have the following natural questions.

* "I agree that the phenomenon (High TTL does not necessarily indicate a better posterior approximation) is observed in Section 4. However, it is better to explain more why it occurs from more broader viewpoints. According to the Bernstein–von Mises theorem, as the sample size approaches infinity and the model is correctly specified, the posterior should converge to the true parameter. Therefore, when the sample size is sufficiently large, I guess this phenomenon should not occur? So, is it due to an insufficient sample size?

* I mentioned the Bernstein–von Mises theorem. However, the assumptions of this theorem are unmet in complex models like in the case of Bayesian deep neural networks, although I believe they are satisfied in Section 3. This could also contribute to the observed phenomenon.

* In this context, the author evaluates the difference between the exact posterior and the approximated posterior using the Wasserstein distance. However, what if we employ alternative metrics, such as the KL distance? Perhaps this is also linked to the underlying cause of the observed phenomenon?

Summary: Primarily, I aim to convey that this phenomenon would arise from several intertwined, overarching factors. Readers would benefit from a clearer explanation of the specific causes behind this phenomenon. Could you succinctly incorporate these reasons?

### In Section 5

I concur with the primary assertion that TTL may not be appropriate depending on the objectives of our analysis. However, the subsequent recommendations in the paper are rather ambiguous. It remains unclear how these suggestions can be implemented in practical terms. Readers would appreciate more explicit and actionable recommendations. Here are some specific comments.

* The author argues that if our analysis aims to approximate the posterior, TTL might not be suitable. In that case, what types of experiments should we report? Should we calculate the Wasserstein distance, as the author did in toy experiments with correctly-specified models?

* The author suggests that depending on their goals, it might be better to use a different scoring rule than KL. However, I believe analysts often lack a clear intuition about which divergence measure to use, which is why people typically resort to KL divergence. In such cases, what should analysts do to compare methods?

   *  I believe another practical and natural suggestion we can offer is to include results with various divergence measures as a robust sanity check.

* As the author pointed out, in cases of model misspecification (which is common in the real world), TTL becomes impractical. Similarly, when dealing with complex data, calculating the ground truth posterior is impossible. What approach should analysts take to compare methods in such situations?

   * In general, there may be limited options. However, in specific domains like images, if our objective is to obtain samples of "natural images," we can employ scoring functions that quantify the naturalness of images. Many variations of such functions are commonly utilized as evaluation metrics for natural images. It might be worthwhile to highlight such cases where objectives are well-defined.

**Strengths And Weaknesses:**

## Strengths

The paper is overall very clear and organized. I vote for the acceptance.

## Weakness

Explanation of some parts would be lacking.

---

> ### Author Response · Authors · 2023-11-05
> **Response to MA33**
>
> **Readers would appreciate more high-level explanations for this phenomenon, not necessarily specific to each instance:**
>
> We thank the reviewer for the suggestion. We have added a general discussion focused on intuition in a new subsection, Section 3.4, as well as a cartoon illustration of the phenomenon (Figure 5).
>
> **According to the Bernstein–von Mises theorem, as the sample size approaches infinity and the model is correctly specified, the posterior should converge to the true parameter. Therefore, when the sample size is sufficiently large, I guess this phenomenon should not occur? So, is it due to an insufficient sample size:**
>
>  We appreciate the referee’s raising this possibility.
> We agree that (1) in a well-specified model, (2) with sufficient regularity conditions, and (3) with a large enough training sample, we can expect that the posterior will concentrate near the true parameter value, and we might further expect the posterior predictive to be close to the true data-generating distribution. Even if this full argument holds true, though, we note that it has two major issues in practice. First, in every real data analysis problem, we expect the model to be mis-specified. And second, we are interested in Bayesian posterior approximation precisely when the posterior distribution is non-trivial. If the practitioner knows in advance that they have sufficient training data to pin down the true parameter with essentially no uncertainty, it's not clear that the use of Bayesian inference (or approximate Bayesian inference) is necessary.
>
> We have added a discussion of this interesting point to the manuscript as part of our new Section 3.4.
>
> **Alternate notions (instead of 2-Wasserstein distance) of posterior closeness:**
> The reviewer wonders if anything substantive changes if we use KL to measure the distance with the true posterior. We can confirm that nothing substantive changes; to illustrate, we have added an additional figure in the appendix (Figure 11) that is analogous to the plots in Figures 1 and 4 but uses the reverse KL divergence as opposed to the 2-Wasserstein distance. We observe the same behavior between TLL and KL as with TLL and the 2-Wasserstein distance: namely, trends in TLL need not reflect trends in the divergences, and the model with best TLL does not  correspond to the model with best divergence. Moreover, Figures 9 and 10 plot the difference of marginal standard deviations of the parameters of interest between an approximation and the exact posterior vs. TLL and illustrate similar discrepancies. Together, these examples highlight that the trends in TLL need not be meaningfully related to a variety of sensible measures of posterior closeness.
>
> We hope that our new Section 3.4 makes clear why we expect this discrepancy, regardless of the divergence used.
>
> **I concur with the primary assertion that TLL may not be appropriate depending on the objectives of our analysis. However, the subsequent recommendations in the paper are rather ambiguous:**
> Thank you for highlighting that we could provide more concrete recommendations and for your thoughtful suggestions. We have updated Section 5 in the paper to more thoroughly discuss recommendations incorporating the spirit of many of your suggestions.

---

### Author Response · Authors · 2023-11-05
**To all reviewers**

We thank the reviewers for their thoughtful comments. We have updated the paper to reflect many of the suggested changes, provided additional details and derivation in both the main text and the appendix, and have uploaded the code to reproduce the experiments in the paper.

---

### Decision · Action_Editor_vuFb · 2023-12-19

**Recommendation:** Accept as is

**Comment:**

The paper studies the use of test log-likelihood to assess Bayesian inference algorithms or to compare forecast accuracy of different models. It provides examples and theory that well demonstrate what test log-likelihood does and does not measure.

While not proposing new methods, the reviewers believe that the paper will be useful for parts of the community either as a learning tool or as useful reference.

The reviewers have raised a number of concerns during the review cycle. The majority of them are satisfied that the revision has addressed the concerns. Moreover, the concerns raised by reviewer m596 are discussed, while not as fully as one could potentially do, at least sufficiently in Appendix C.

**Audience:**

The paper will be of interest some of the TMLR's audience, either as a learning tool or as a reference.

**Claims And Evidence:**

The evidence provided in the paper is convincing and clear.